# FUSING REWARDS AND PREFERENCES IN REINFORCEMENT LEARNING

## ABSTRACT

We present Dual-Feedback Actor (DFA), a reinforcement learning algorithm that fuses both individual rewards and pairwise preferences (if available) into a single update rule. DFA uses the policy's log-probabilities directly to model the preference probability, avoiding a separate reward-modeling step. Preferences can be provided by human-annotators (at state-level or trajectory-level) or be synthesized online from Q-values stored in an off-policy replay buffer. Under a Bradley–Terry model, we prove that minimizing DFA's preference loss recovers the entropy-regularized Soft Actor-Critic (SAC) policy. Our simulation results show that DFA trained on generated preferences matches or exceeds SAC on six control environments and demonstrates a more stable training process. With only a semi-synthetic preference dataset under Bradley-Terry model, our algorithm outperforms reward-modeling reinforcement learning from human feedback (RLHF) baselines in a stochastic GridWorld and approaches the performance of an oracle with true rewards.

## 1 INTRODUCTION

Over the past decade, Reinforcement Learning (RL) has achieved remarkable success across a wide range of applications, including video games (Knox & Stone, 2008; Warnell et al., 2018), recommendation systems (Kohli et al., 2013; Zeng et al., 2016), and autonomous driving (Kiran et al., 2021). RL focuses on how agents make decisions while interacting with dynamic, changing environments. At each time step, an agent chooses an action based on its current state and receives a reward that indicates how good that action was. The goal is to learn a policy that maximizes the total reward accumulated over time. In traditional RL, the reward function is usually manually designed by experts to guide the agent's behavior toward desired outcomes. However, crafting such a function is a challenging and often ambiguous task (Ng et al., 2000).

To overcome the limitations of hand-engineered rewards, Reinforcement Learning from Human Feedback (RLHF) has emerged as a compelling alternative, particularly in the fine-tuning of large language models (LLMs) (Christiano et al., 2017; Stiennon et al., 2020; Ouyang et al., 2022). RLHF bypasses manual reward specification by inferring a reward model from human preferences over trajectory pairs. This reward model then guides policy optimization using standard RL algorithms. Despite its empirical successes, RLHF methods relying on reward inference, face significant practical and theoretical challenges, including reward model misspecification, overfitting, distribution shift, and non-identifiability of reward functions (Zhu et al., 2024; Casper et al., 2023). Moreover, the reward inference step introduces additional complexity and often requires large volumes of annotated data.

To simplify the pipeline and avoid reward inference, in the context of language modeling, Direct Preference Optimization (DPO) has recently been proposed as a direct approach to exploit human preferences (Rafailov et al., 2023). Thanks to a closed-form expression of the optimal policy under a Bradley–Terry preference model, DPO avoids estimating the reward function. Although DPO has shown promising results in fine-tuning large language models, its loss formulation tends to induce deterministic policies and is susceptible to mode collapse (Azar et al., 2024; Sharifnassab et al., 2024). Moreover, the existing theory for DPO only covers contextual bandits or MDPs with deterministic transitions (Rafailov et al., 2023; 2024). As a result, directly applying DPO (or methods suggested in Guo et al. (2024); Xie et al. (2024)) in general reinforcement learning settings is suboptimal, where effective exploration is critical for policy improvement in stochastic MDPs (Zhang & Ying, 2024).

More recently, ZPG (Zhang & Ying, 2024) suggested an RLHF approach that does not rely on a reward model and is designed for non-deterministic MDPs. However, as the authors acknowledged, the algorithm lacks a strategic exploration mechanism. Furthermore, it relies on trajectory-level preference comparisons and performs on-policy updates, hence previously collected data are not reused.

In this work, we introduce Dual-Feedback Actor (DFA), a reinforcement learning algorithm that works for stochastic MDPs and unifies scalar rewards and preference-based feedback into a single, principled policy update rule. Unlike many prior approaches in RLHF that infer a separate reward model from preferences, DFA directly incorporates preferences into the policy optimization objective using the policy's log-probabilities and retains Soft Actor-Critic (SAC)-style entropy-driven exploration. The main contributions are as follows:

- Our approach offers dual compatibility with both rewards and preferences. When numerical rewards are available, the agent updates its Q-networks and incorporates preference-based learning by synthesizing preferences from Q-values. This dual approach allows the agent to use reward signals while maintaining flexibility to incorporate human feedback, especially in settings where rewards are sparse or absent.

- Our approach can be used not only in on-policy manner but also in off-policy manner, which enables more sample-efficient learning by reusing past experiences stored in a replay buffer. This is particularly valuable for hierarchical RL applications where sample efficiency is needed to train the policy of each layer.

- Under the assumptions stated in Section 5, we prove that minimizing DFA's preference loss recovers the entropy-regularized SAC solution, formally bridging preference optimization and entropy-regularized RL. Consequently, DFA inherits SAC's entropy-driven exploration, maintaining diverse action sampling even when it learns solely from preferences.

- Experimental results in Section 6 show that DFA consistently matches or outperforms both reward-based and preference-based baselines on six control tasks and a stochastic GridWorld, while yielding a more stable training process.

## 2 RELATED WORK

There are two dominant paradigms for incorporating human feedback in reinforcement learning. The first relies on reward modeling: These methods first fit a scalar reward (or value) prediction model from preference data and then treat this learned reward model as the surrogate reward for standard policy optimization. This two-stage pipeline was introduced in (Christiano et al., 2017), Schoenauer et al. (2014), and later scaled to large language models by Ziegler et al. (2019), Stiennon et al. (2020), and Ouyang et al. (2022). The second relies on direct policy optimization: These algorithms bypass an explicit reward model and update the policy parameters solely from preference comparisons (Wilson et al., 2012; Busa-Fekete et al., 2014; Akrour et al., 2011).

For reward-modeling approaches, several works (Saha et al., 2023; Zhu et al., 2023; Wu & Sun, 2023) consider linearly parameterized reward models and characterize the error bounds of the estimated parameters, and prove that subsequent reward-based RL can tolerate small errors in rewards. Zhan et al. (2023) extend this analysis to more general reward function classes under some conditions. These analyses have been extended to direct policy optimization approaches in Xu et al. (2020); Chen et al. (2022); Zhang et al. (2024).

In the context of language modeling, DPO (Rafailov et al., 2023) provides a direct approach to aligning language models with human preferences by optimizing a policy to maximize the likelihood of preferred responses over nonpreferred ones, eliminating the need for an explicit reward model. SPO (Sharifnassab et al., 2024) optimizes model output directly over a preference dataset through the natural conditional probability of the preferred responses over nonpreferred ones. Similar approaches have also been explored in this literature (Xu et al., 2024; Ethayarajh et al., 2024; Hong et al., 2024; Park et al., 2024; Hong et al., 2024; Meng et al., 2024; Li et al., 2025). RLHF has also been studied in other aspects. For example, the framework in Swamy et al. (2024) casts RLHF as a two-player zero-sum game. However, they still estimate the rewards (and subsequently apply PPO, TRPO, or SAC) based on a constantly updated queue of recent rollouts, which can cause data staleness issues.

Recent work, Xie et al. (2024), inspired by DPO, combines DPO with optimistic exploration to design XPO in the function approximation regime with provable convergence. ZPG (Zhang & Ying, 2024) aims to address RLHF without relying on a reward model and is designed for non-deterministic MDPs. However, it lacks an exploration mechanism, which is essential for general RL applications. Although previous work brought advancement in several aspects, existing algorithms are rarely benchmarked (theoretically and experimentally) against strong reward-based baselines such as SAC.

## 3 PRELIMINARIES

In this section, we introduce the notation for RL, RLHF, and review the DPO objective (Rafailov et al., 2023). We model the environment as a finite–horizon Markov Decision Process (MDP). An MDP can be represented as a tuple $\mathcal{M} = \langle \mathcal{S}, \mathcal{A}, P, R, \gamma, p_0 \rangle$, where $\mathcal{S}$ and $\mathcal{A}$ are state space and action space, respectively. The conditional probability of transition from state $s$ to $s'$ with action $a$ is denoted by $P(s'|s, a)$. The probability distribution over the initial state $s_0$ is denoted by $p_0(s_0)$. The parameter $\gamma \in (0, 1)$ denotes the discount factor. At each time step $t$, $r(s_t, a_t)$ returns the reward of taking action $a_t$ in the state $s_t$. Actions are chosen according to the policy $\pi$ where $\pi(a|s)$ is the probability of taking action $a$ for a given state $s$. Here, we assume that the policy is parameterized with a vector $\theta \in \mathbb{R}^d$ and use shorthand notation $\pi_\theta$ for $\pi_\theta(a|s)$. For a given time horizon $H$, we define $\tau = (s_0, a_0, \cdots, s_{H-1}, a_{H-1})$ as a sequence of state-action pairs called a trajectory. $R(\tau)$ is a function that returns the discounted accumulated reward of each trajectory as follows: $R(\tau) := \sum_{h=0}^{H-1} \gamma^h r(s_h, a_h)$ where $\gamma \in (0, 1)$ is the discount factor.

Given a policy $\pi$, the *state-value function* and the *action-value function (or Q-function)* are

$$V^\pi(s) = \mathbb{E}_{\tau \sim \pi}\Big[\sum_{t=0}^{H-1} \gamma^t\, r(s_t, a_t) \,\Big|\, s_0 = s\Big], Q^\pi(s,a) = \mathbb{E}_{\tau \sim \pi}\Big[\sum_{t=0}^{H-1} \gamma^t\, r(s_t, a_t) \,\Big|\, s_0 = s,\ a_0 = a\Big].$$

**Classical RLHF feedback setting (Christiano et al., 2017).** Let $\mathcal{M} = \langle \mathcal{S}, \mathcal{A}, P, R, \gamma, p_0 \rangle$ be the finite-horizon MDP where the true reward $r(s, a)$ is *hidden*. Hence, we ask humans to compare trajectories and form the preference dataset

$$\mathsf{D} = \big\{(\tau_k^+, \tau_k^-)\big\}_{k=1}^K, \qquad \tau_k^+ \succ \tau_k^-,$$

where $K$ is the total number of pairs, and $\tau_k^+$ is *preferred* to $\tau_k^-$ which is denoted as $\tau_k^+ \succ \tau_k^-$. We define a parametric function $r_\phi : \mathcal{S} \times \mathcal{A} \to \mathbb{R}$ to *approximate* the latent reward. For any trajectory $\tau = (s_0, a_0, \ldots, s_{H-1}, a_{H-1})$, we define the model return as:

$$R_\phi(\tau) = \sum_{h=0}^{H-1} \gamma^h\, r_\phi(s_h, a_h).$$

The parameters $\phi$ are learned by maximum likelihood under the Bradley–Terry model (Bradley & Terry, 1952), which is equivalent to minimizing the following loss:

$$\mathcal{L}(\phi) = -\,\mathbb{E}_{(\tau^+, \tau^-) \sim \mathsf{D}}\big[\log \sigma\big(R_\phi(\tau^+) - R_\phi(\tau^-)\big)\big], \qquad \sigma(z) = \frac{1}{1 + e^{-z}}.$$

Let $\hat{r}_\phi$ be the estimated reward function. Next, with $\hat{r}_\phi$ fixed, a policy-gradient method such as PPO (Schulman et al., 2017) or SAC (Haarnoja et al., 2018) updates $\pi_\theta$ to maximize

$$J(\theta) = \mathbb{E}_{\tau \sim \pi_\theta}\big[\hat{R}_\phi(\tau)\big].$$

A well-known drawback on this two-stage pipeline is its sensitivity to noise, and any overfitting in $\hat{r}_\phi$ propagates directly to the final policy updates (Casper et al., 2023).

**DPO in language models (Rafailov et al., 2023)** In language models, a *state* is the text prefix (or prompt) $x$, and an *action* is the response $y$ produced by the model (call it continuations). Annotators make a choice among two full continuations $(y^+, y^-)$ sampled from the same prompt, giving the preference dataset

$$\mathsf{D} = \big\{(x_k, y_k^+, y_k^-)\big\}_{k=1}^K, \qquad y_k^+ \succ y_k^-.$$

Let $\pi_{\text{ref}}$ be the frozen base model (e.g. a pre-trained GPT checkpoint). For a prompt $x$ and two candidate continuations $y^+, y^-$, define the log-probability gap:

$$\Delta_{x,y^+,y^-}(\theta) = \left[\log \pi_\theta(y^+ \mid x) - \log \pi_\theta(y^- \mid x)\right] - \left[\log \pi_{\text{ref}}(y^+ \mid x) - \log \pi_{\text{ref}}(y^- \mid x)\right].$$

$\Delta_{x,y^+,y^-}(\theta)$ captures how much more the new model prefers the chosen continuation over the rejected one, *relative* to the base model. DPO then minimizes

$$\mathcal{L}_{\text{DPO}}(\theta) = - \mathbb{E}_{(x,y^+,y^-)\sim\mathsf{D}}\left[\log \sigma\big(\alpha\,\Delta_{x,y^+,y^-}(\theta)\big)\right], \qquad \sigma(z) = \tfrac{1}{1+e^{-z}},\; \alpha > 0.$$

Minimizing $\mathcal{L}_{\text{DPO}}$ pushes the new model toward the preferred continuation, while limiting it to the safe behavior of $\pi_{\text{ref}}$. The absence of a separate reward model in DPO removes a major source of overfitting or noisy evaluations of the reward modeling. However, DPO assumes a Bradley-Terry choice model to derive its loss function, and this loss tends to produce near-deterministic models. This reduced diversity makes DPO prone to mode collapse (Azar et al., 2024).

# 4 METHODS

In this section, we introduce our *Dual-Feedback Actor* (DFA). In order to describe the DFA algorithm, we first introduce the state-wise feedback setting as follows:

**State-wise feedback.**   In this setting, the annotator does *not* compare full trajectories. Instead, at a given state $s_k$, the annotator sees two actions, marks the winner $a_k^+$ over the loser $a_k^-$. Then, the following preference dataset is formed:

$$\mathsf{D} \;=\; \big\{(s_k, a_k^+, a_k^-)\big\}_{k=1}^{K}, \qquad a_k^+ \succ a_k^-,$$

where $a^+$ is *preferred* to $a^-$ at state $s_k$. In the subsections below, we first consider the case where the agent learns only from state-wise human comparisons. Second, we show how to synthesize preferences from numerical rewards when they are available. Finally, we extend DFA to trajectory-based comparisons.

## 4.1 LEARNING WITH ONLY STATE-WISE PREFERENCES

Assume we have collected a set of preference comparisons

$$\mathsf{D} \;=\; \big\{(s_k, a_k^+, a_k^-)\big\}_{k=1}^{K}, \qquad a_k^+ \succ a_k^- \;.$$

Unlike classical RLHF, *we do not assume an underlying Bradley–Terry reward model*. Instead, we rely on the policy's log-probabilities to model the preference probability directly. For any pair $(s, a^+, a^-)$ we define the *preference probability* produced by the current policy $\pi_\theta$ as

$$P_\theta\big(a^+ \succ a^- \mid s\big) = \frac{\pi_\theta(a^+ \mid s)^\alpha}{\pi_\theta(a^+ \mid s)^\alpha + \pi_\theta(a^- \mid s)^\alpha}, \qquad \alpha > 0. \tag{1}$$

The exponent $\alpha$ controls the uncertainty assigned to the policy's output: $\alpha \to 0$ yields a nearly uniform (high-entropy) choice, while $\alpha \to \infty$ approaches a hard winner–takes–all rule.

The negative log-likelihood equation 1 gives the state-wise preference loss:

$$\mathcal{L}_{\text{pref}}(\theta) \;=\; - \mathbb{E}_{(s,a^+,a^-)\sim\mathsf{D}}\big[\log P_\theta\big(a^+ \succ a^- \mid s\big)\big]. \tag{2}$$

Minimizing $\mathcal{L}_{\text{pref}}$ directly increases the probability that $\pi_\theta$ selects the human-preferred action, without introducing auxiliary reward networks or relying on any latent utility assumptions [1]. Note that equation 2 can be reformulated as follows:

---

[1]Eq. equation 2 is identical to the *preference loss* $\mathcal{L}_{\text{pref}}^\alpha$ used in Soft Preference Optimization (SPO) (Sharif-nassab et al., 2024). Although DFA adopts the same logistic pairwise-loss form, the similarity ends there. In SPO, the same term is combined with a global KL regularizer $\mathrm{D}_{\text{KL}}(\pi_\theta \,\|\, \pi_{\text{ref}})$, whereas here we study the stand-alone preference part and show that, under some assumptions, it aligns the policy with the entropy–regularized RL solution (Theorem 5.2). Moreover, SPO is in the context of LLMs and is designed for an offline setting. DFA targets stochastic MDPs, supports off-policy replay, preserves SAC-style entropy exploration with theoretical analysis, and unifies numeric rewards with preferences. Synthesizing preferences, as explained in Section 4.2, is another key innovation in DFA that allows for online settings in RL.

$$\mathcal{L}_{\text{pref}}(\theta) = -\mathbb{E}_{(s,a^+,a^-)\sim\text{D}}\Big[\log\sigma\Big(\alpha\big(\log\pi_\theta(a^+|s) - \log\pi_\theta(a^-|s)\big)\Big)\Big]$$

In simulated environments or settings where numerical rewards are accessible, it is possible to synthesize preference data from these rewards or their proxies, such as Q-values. Our approach, introduced in the next section, is particularly useful when integrating preference-based learning into an agent's training loop, even when direct human feedback is unavailable or insufficient. Our method fuses numerical rewards and preference data by synthesizing preferences from numerical rewards.

## 4.2 SYNTHESIZING PREFERENCES FROM NUMERICAL REWARDS

We use Q-values as a proxy to create preference pairs, enabling online preference generation during policy updates without explicitly constructing full trajectory segments. Estimating Q-values can be done through any method in the literature, and is particularly relevant in off-policy methods such as SAC, where a replay buffer stores past experiences as tuples $(s_t, a_t, r_t)$, where $s_t$, $a_t$, and $r_t$ are state, action and reward at time $t$, respectively.

Our approach works as follows: For a batch of states $\{s_i\}_{i=1}^{N}$ sampled from the replay buffer, we generate two candidate actions to form preference pairs: The first action, denoted by $a_i$, corresponds to the action originally taken in state $s_i$ as stored in the replay buffer. This action reflects the historical behavior of the agent at the time the state was visited. The second action, denoted by $a_i'$, is obtained from the replay buffer by identifying the action associated with the nearest state to $s_i$ (denote it with $s_i'$)[2]. For both actions, we compute their respective Q-values. The action with the higher Q-value is designated as the preferred action $a_i^+$, while the other is labeled as the rejected action $a_i^-$:

$$\text{If } Q(s_i, a_i) > Q(s_i, a_i'), \qquad \text{then } (a_i^+, a_i^-) = (a_i, a_i'),$$

$$\text{else } (a_i^+, a_i^-) = (a_i', a_i).$$

This process effectively synthesizes preference data in the form of state-action pairs $\text{D}^{\text{Syn}} = \{(s_i, a_i^+, a_i^-)\}_{i=1}^{N}$ for each batch. The loss in equation 2, is then calculated over the states $\{s_i\}_{i=1}^{N}$ and using their associated preferred and rejected actions. Specifically, the loss encourages the policy to assign higher probability to preferred actions over rejected ones, scaled by the parameter $\alpha$

$$\mathcal{L}_{\text{pref}}^{\text{Syn}}(\theta) = -\mathbb{E}_{(s_i,a_i^+,a_i^-)\sim\text{D}^{\text{Syn}}}\Big[\log\Big(\sigma\big(\alpha\big(\log\pi_\theta(a_i^+|s_i) - \log\pi_\theta(a_i^-|s_i)\big)\big)\Big)\Big]$$

where $\sigma(\cdot)$ is the sigmoid function. Figure 1 gives a high-level schematic of our methodology.

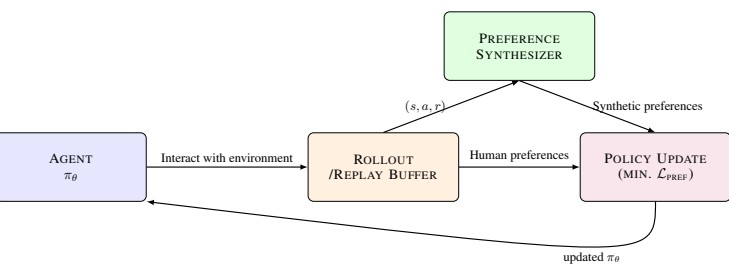

Figure 1: Data flow in DFA. The agent executes its current policy $\pi_\theta$ and stores the transitions (may include reward-based transitions or human-annotated preferences). If reward-based transitions are available, the *Preference Synthesizer* can convert them into synthetic preference pairs. This process can be done in either an on-policy or off-policy fashion. Both human and synthetic preferences can be used in *Policy-Update*, which minimizes the preference loss $\mathcal{L}_{\text{pref}}$ and outputs an improved policy.

---

[2]In our experiments, we compute the Euclidean distance between $s_i$ and all states in the buffer, select the closest state $s_i'$, and retrieve its corresponding action $a_i'$.

### 4.3 EXTENSION OF THE LOSS TO TRAJECTORY-BASED COMPARISONS

State-wise comparisons can be easy to collect (for instance, a single frame rather than a full video in video games) and give richer training signals, but one may prefer to rank the whole trajectories (Christiano et al., 2017; Zhang & Ying, 2024), hence, we extend DFA to accept trajectory-level preferences as well. For trajectory-level comparisons, we store pairs

$$\mathsf{D}^{\text{traj}} = \big\{(\tau_k^+, \tau_k^-)\big\}_{k=1}^K, \qquad \tau = (s_1, a_1, \ldots, s_T).$$

The policy assigns a likelihood to any full trajectory as follows: $\pi_\theta(\tau) = \prod_{t=1}^T \pi_\theta(a_t \mid s_t)$. The preference probability is the same as before, but now in terms of trajectory likelihoods:

$$P_\theta^{\text{traj}}(\tau^+ \succ \tau^-) = \frac{\pi_\theta(\tau^+)^\alpha}{\pi_\theta(\tau^+)^\alpha + \pi_\theta(\tau^-)^\alpha}, \qquad \alpha > 0. \tag{3}$$

The negative log-likelihood of equation 3 gives trajectory-based preference loss:

$$\mathcal{L}_{\text{pref}}^{\text{traj}}(\theta) = - \mathbb{E}_{(\tau^+, \tau^-) \sim \mathsf{D}^{\text{traj}}}\Big[\log P_\theta^{\text{traj}}(\tau^+ \succ \tau^-)\Big]. \tag{4}$$

## 5 THEORETICAL ANALYSIS

In this section, we show that, under Bradley–Terry model on the soft optimal $Q$–function, minimizing our preference loss is equivalent to recovering the optimal policy for entropy–regularized reinforcement learning (Haarnoja et al., 2017). Concretely, we analyze the tabular setting for the state-wise preferences and identify its unique minimizer. This establishes the equivalence of preference optimization and entropy-regularized RL. We should emphasize that the BT model is not a requirement of DFA Algorithm; it is only used to derive Theorem 5.2 in the following. A trajectory-wise analysis and its connection to the state-wise analysis, is provided in Appendix B.

**Assumption 5.1** (Bradley–Terry preferences on the *soft*-optimal $Q$-function). Let $Q^\star : \mathcal{S} \times \mathcal{A} \to \mathbb{R}$ be the soft-optimal state-action value function of the MDP, i.e.,

$$Q^\star(s, a) = \max_\pi \mathbb{E}\Big[\sum_{t=0}^\infty \gamma^t\big(r(s_t, a_t) + \lambda\,\mathcal{H}(\pi(\cdot \mid s_t))\big) \;\Big|\; s_0 = s,\, a_0 = a\Big],$$

where $\mathcal{H}(.)$ is the entropy function and $\lambda$ is the entropy coefficient. Assume that there exists a parameter $\beta > 0$ such that, for every $s \in \mathcal{S}$ and any $a, b \in \mathcal{A}$,

$$P^\star(a \succ b \mid s) = \sigma\Big(\beta\,[\,Q^\star(s, a) - Q^\star(s, b)\,]\Big), \qquad \sigma(z) = \frac{1}{1 + e^{-z}}.$$

**Theorem 5.2** (Preference loss recovers the optimal policy). *Fix a state $s \in \mathcal{S}$ and abbreviate $Q_a^\star := Q^\star(s, a)$. Suppose that Theorem 5.1 holds. Under uniform sampling of ordered pairs $(a, b) \sim \text{Unif}(\mathcal{A}^2)$ and the tabular full-support parameterization $\ell_a = \log \pi(a \mid s)$ ($\sum_a e^{\ell_a} = 1$, $e^{\ell_a} > 0$), consider the preference loss*

$$\mathcal{L}(\boldsymbol{\ell}) = -\frac{1}{|\mathcal{A}|^2} \sum_{(a,b) \in \mathcal{A}} P^\star(a \succ b \mid s) \log \sigma\big(\alpha(\ell_a - \ell_b)\big), \qquad \alpha > 0.$$

*This loss is strictly convex on the set of full-support policies and is minimized uniquely at*

$$\pi_\star(a \mid s) = \frac{\exp\big(\frac{\beta}{\alpha}\,Q_a^\star\big)}{\sum_{a' \in \mathcal{A}} \exp\big(\frac{\beta}{\alpha}\,Q_{a'}^\star\big)}. \tag{5}$$

*Furthermore, this Gibbs distribution coincides with the global maximizer of the entropy-regularized RL (or SAC objective) when $\lambda = \alpha/\beta$:[3]*

$$\max_\pi \mathbb{E}_\pi\Big[\sum_{t=0}^\infty \gamma^t\big(r(s_t, a_t) + \lambda\,\mathcal{H}(\pi(\cdot \mid s_t))\big)\Big]. \tag{6}$$

---

[3]All proofs are provided in the Appendix.

Theorem 5.2 states that, when human (or synthetic) comparisons follow a Bradley-Terry model whose latent utility equals the ground truth $Q^\star$, the preference loss is perfectly aligned with the entropy-regularized control objective (Haarnoja et al., 2017). The optimizer equation 5 is soft-max policy whose inverse temperature is the ratio $\beta/\alpha$: The parameter $\beta$ captures how consistently the annotator prefers higher-value actions, while the parameter $\alpha$ adjusts the learner's uncertainty. In particular, setting $\lambda = \alpha/\beta$ recovers the SAC trade-off between exploitation (large $\beta$) and exploration (large $\alpha$) (Haarnoja et al., 2018).

*Remark* 5.3. If the Bradley-Terry assumption holds for any *arbitrary* soft state-action value function, for instance, the current critic estimate $Q_k(s,a)$ in SAC (Haarnoja et al., 2018). Then Theorem 5.2 implies that the preference loss is minimized by

$$\pi_{k+1}(a \mid s) = \frac{\exp\!\big(\frac{\beta}{\alpha}\,Q_k(s,a)\big)}{\sum_{a'} \exp\!\big(\frac{\beta}{\alpha}\,Q_k(s,a')\big)}.$$

This update is *exactly* the policy-improvement step in SAC that maximizes the entropy-regularized objective

$$\max_{\pi} \Big\{ \mathbb{E}_{a\sim\pi}\big[Q_k(s,a)\big] + \lambda\,\mathcal{H}\big(\pi(\cdot \mid s)\big) \Big\}.$$

Hence, as the critic converges ($Q_k \to Q^\star$), repeated minimization of the preference loss yields the soft-optimal SAC policy. Therefore, preference learning can be viewed as performing policy improvement in SAC, but driven solely by comparative feedback.

*Remark* 5.4. The assumption that $(a,b) \sim \mathrm{Unif}(\mathcal{A}^2)$ in Theorem 5.2 is made for simplicity of analysis. In practice, one can approximate this condition by drawing a mini-batch of ordered pairs at each update and down-sampling (or re-weighting) each pair by the inverse of its frequency in the batch; This produces a uniform sub-sampled action pairs required by the theorem.

## 6 EXPERIMENTAL RESULTS

In this section, we benchmark DFA against prior work. The complete code is provided in the supplementary material. We first compare DFA with the reward-based baseline SAC (hence, we have to synthesize preferences following Section 4.2), and then against recent preference-based methods.

### 6.1 COMPARISON WITH SAC VIA SYNTHETIC PREFERENCES

In this section, we evaluate the proposed algorithm (DFA) and compare it with related work on six control tasks in MuJoCo (Todorov et al., 2012), a physics simulator known for fast and accurate simulations in areas such as robotics, biomechanics, and graphics. Since published benchmarks (e.g., OpenAI SpinningUp) consistently identify SAC as the strongest baseline on many environments, we compare DFA exclusively with SAC. We briefly explain the six environments we consider in Appendix C.

In Figure 2, we monitor the average episode return versus system probes, which represents the total number of environment interactions. In this experiment, DFA continually generates synthetic preference pairs from numerical rewards following 4.2. The underlying RL settings and replay buffer are identical to those of SAC. Both DFA and SAC run for $10\times10^6$ system probes across 5 different random seeds. We use mini-batch size 256 for both algorithms. A new preference batch of size $N = 256$ is created during every gradient step. Figure 2 shows that DFA matches or exceeds SAC on Walker2d, Hopper, Swimmer, and Humanoid. In MountainCarContinuous, we could not find SAC settings that produced learning, a problem others have reported as well.[4] DFA, in contrast, learned a good policy on this task with the same range of hyperparameters used for the other environments.

Interestingly, DFA's learning curves are noticeably smoother, while SAC exhibits significant fluctuations. We attribute this stability to the synthesized preference pairs, which are constructed according to Section 4.2, and appear to act as an implicit denoising regularizer. We note that reducing the learning rate or adjusting other hyperparameters to avoid fluctuations for SAC resulted in lower average returns, thus, we maintained the higher learning rate configuration to ensure fair comparison.

---

[4]https://github.com/rail-berkeley/softlearning/issues/76

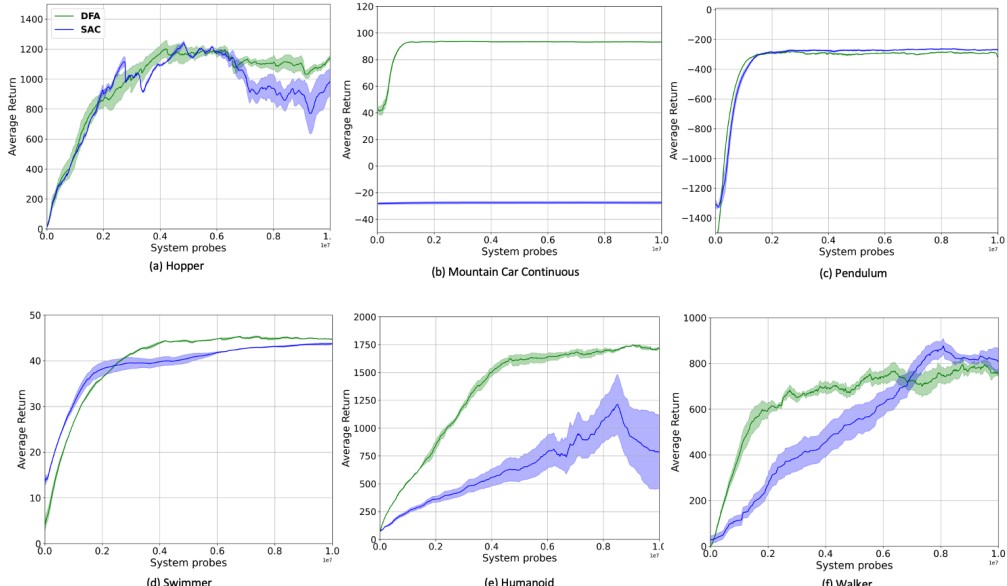

Figure 2: DFA (green) vs. SAC (blue) on the six MuJoCo control tasks. DFA matches or exceeds SAC and shows smoother training. The solid line is the mean episode return, and the shaded region shows an 90% confidence interval over 5 seeds.

These results confirm the claim of Theorem 5.2: *once we use preference data aligned with the optimal Q-values, numerical rewards can be dropped without losing performance.* This unifies reward-free human alignment and reward-based RL under a single log-likelihood objective.

## 6.2 COMPARISON WITH RM METHODS

In this section, we evaluate our DFA algorithm against traditional reward modeling approaches in the context of learning from human preferences. While the previous section demonstrated DFA's effectiveness with generated preferences derived from numerical rewards, here we focus on the more challenging scenario where only human comparative feedback is available, without access to ground-truth rewards.

We conduct experiments in a stochastic GridWorld environment, which provides a controlled testbed for preference-based learning (Zhang & Ying, 2024). In this environment, the agent starts at the center of the grid and can take four actions: up, down, left, or right. The environment includes the following aspects: (1) To build the ground play, a coin is flipped for each cell, and if heads, a reward sampled from $\mathcal{N}(0,1)$ is placed in that cell; (2) While the agent is moving, with probability $0.4$, the chosen action is reversed (e.g., "up" becomes "down"). Each episode has a fixed horizon of 20 steps, and the agent's goal is to maximize the cumulative reward collected. This environment is particularly suitable for preference-based learning evaluation as it combines stochastic dynamics with a non-trivial reward structure that requires exploration.

To simulate human preferences, we simulate a panel of annotators who provide comparative feedback between trajectories. Following standard practice in RLHF literature, we model the annotator's preference probability using the Bradley-Terry model: $P(\tau_1 \succ \tau_0) = \sigma(R_1 - R_0)$, where $R_i$ is the cumulative reward of trajectory $\tau_i$ and $\sigma$ is the sigmoid function. For robustness, each preference query aggregates votes from $M$ independent annotators, and the majority vote determines the final preference. This approach simulates the noise and variability in real human feedback while maintaining a consistent underlying reward structure. For more implementation details, please see Appendix C. We compare DFA against the following approaches: **RM+PPO**: A two-stage approach that first learns a reward model from preference data using maximum likelihood estimation, then optimizes a policy using Proximal Policy Optimization (PPO) with the learned reward function. **ZPG (Zhang & Ying, 2024)**: A state-of-the-art RLHF method which estimates the policy gradient from

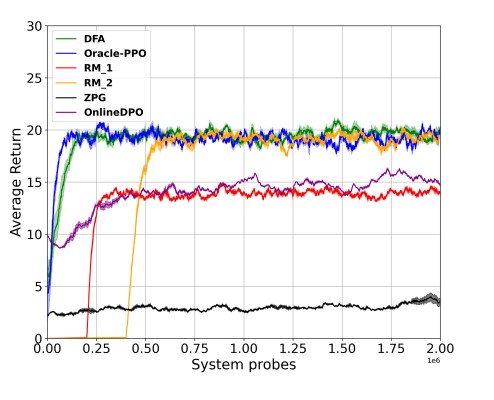 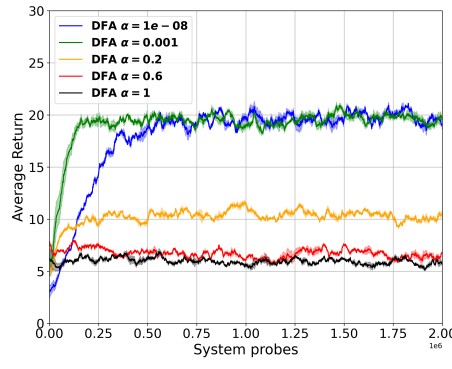

(a) DFA vs. RM + PPO and oracle PPO.

(b) Effect of temperature $\alpha$.

Figure 3: GridWorld results. (a) DFA learns faster and achieves higher rewards than reward-modeling baselines, approaching the oracle that has access to the true reward. (b) Effect of the temperature parameter $\alpha$: a small but not too small value balances exploration and exploitation. Shaded regions denote 90% confidence intervals across 5 random seeds.

preference differences without fitting a reward model. **Oracle-PPO (upper bound)**: PPO directly on the *true* MDP reward $r$ (it is unavailable in practice, but gives an upper bound on the performance.). **OnlineDPO**: We also include the recently-proposed OnlineDPO algorithm (Guo et al., 2024) as a direct-preference baseline.

Figure 3a demonstrates that DFA consistently outperforms reward modeling methods and performs comparably to Oracle-PPO, which has access to the true reward function. In this experiment, we compare against two variants of RM+PPO: RM_1 uses 200k environment steps for training the reward model, while RM_2 uses twice as many samples (400k steps). Despite the increased data budget for RM_2, DFA is still converging faster, highlighting the benefits of avoiding the two-stage pipeline. For ZPG, we used the official repository. Despite our efforts (and implementation tricks such as normalized gradient and gradient clipping), it could not be tuned to outperform the results shown in Figure 3a. We also re-implemented ZPG directly from the paper, closely matching the reported hyperparameters and implementation choices; we report the best observed performance across both implementations. In Figure 3a we use an annotator pool of $M = 500$; runs with smaller $M$ and more complex environments show the same pattern and are included in Appendix C.

Figure 3b highlights the sensitivity of DFA to the parameter $\alpha$. As shown in Figure 3b, setting $\alpha$ too high ($\alpha = 1.0$) gives almost no learning signal, while moderate values in the range $0.2 - 0.6$ yield better results. The best result comes at $\alpha = 0.001$. When $\alpha$ is pushed to very small values (e.g., $10^{-8}$), performance drops again because the policy becomes overly stochastic. These results suggest that $\alpha$ should be small but not too small to balance exploration and exploitation.

# 7 CONCLUSION

Dual-Feedback Actor (DFA) unifies scalar rewards and pairwise preferences in a single loss; when preferences follow a Bradley–Terry model on the optimal soft $Q$-function, this loss recovers the entropy-regularized SAC solution, formally linking reward- and preference-based RL. Empirically, DFA matches or exceeds SAC and outperforms reward-modeling baselines while training more smoothly. The main limitations are the Bradley–Terry assumption, the noise inherited by synthetic preferences from early $Q$ estimates, and the computational cost of finding the nearest state in the replay buffer. The future work can be investigating other assumptions and evaluating DFA on larger, real human-in-the-loop tasks.

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

## A  PROOF OF THE THEOREM 5.2

**Assumption A.1** (Bradley–Terry preferences on the *soft*-optimal $Q$-function). Let $Q^\star : \mathcal{S} \times \mathcal{A} \to \mathbb{R}$ be the soft-optimal state-action value function of the MDP, i.e.,

$$Q^\star(s, a) \;=\; \max_\pi \; \mathbb{E}\Big[ \sum_{t=0}^{\infty} \gamma^t \big( r(s_t, a_t) \;+\; \lambda\, \mathcal{H}(\pi(\cdot \mid s_t))\big)$$

$$\Big|\; s_0 = s, \; a_0 = a\Big],$$

where $\mathcal{H}(.)$ is the entropy function and $\lambda$ is the entropy coefficient. Assume that there exists a parameter $\beta > 0$ such that, for every $s \in \mathcal{S}$ and any $a, b \in \mathcal{A}$,

$$P^\star\big(a \succ b \mid s\big) \;=\; \sigma\Big(\beta\,[\,Q^\star(s, a) - Q^\star(s, b)\,]\Big),$$

$$\sigma(z) \;=\; \frac{1}{1 + e^{-z}}.$$

**Theorem A.2** (Preference loss recovers the optimal policy). *Fix a state $s \in \mathcal{S}$ and abbreviate $Q_a^\star := Q^\star(s, a)$. Suppose that Theorem A.1 holds. Under uniform sampling of ordered pairs $(a, b) \sim \mathrm{Unif}(\mathcal{A}^2)$ and the tabular full-support parameterization $\ell_a = \log \pi(a \mid s)$ ($\sum_a e^{\ell_a} = 1$, $e^{\ell_a} > 0$), consider the preference loss*

$$\mathcal{L}_{pref}(\boldsymbol{\ell}) \;=\; -\,\frac{1}{|\mathcal{A}|^2} \sum_{(a,b) \in \mathcal{A}} P^\star(a \succ b \mid s)\, \log \sigma\big(\alpha(\ell_a - \ell_b)\big),$$

$$\alpha > 0. \tag{7}$$

*This loss is strictly convex on the set of full-support policies and is minimized uniquely at*

$$\pi_\star(a \mid s) \;=\; \frac{\exp\big(\frac{\beta}{\alpha}\, Q_a^\star\big)}{\sum_{a' \in \mathcal{A}} \exp\big(\frac{\beta}{\alpha}\, Q_{a'}^\star\big)}. \tag{8}$$

*Furthermore, this Gibbs distribution coincides with the* global *maximizer of the entropy-regularized RL (or SAC objective) when $\lambda = \alpha/\beta$:*

$$\max_\pi \; \mathbb{E}_\pi\Big[\sum_{t=0}^{\infty} \gamma^t \big( r(s_t, a_t) + \lambda\, \mathcal{H}(\pi(\cdot \mid s_t))\big)\Big]. \tag{9}$$

*Proof.* Because the policy is tabular, we fix the state $s$, and introduce the log-policy vector $\boldsymbol{\ell} = (\ell_a)_{a \in \mathcal{A}}$. Define the policy and the Bradley–Terry probabilities as follows:

$$P_{ab}(\boldsymbol{\ell}) := \sigma\big(\alpha(\ell_a - \ell_b)\big), \qquad P_{ab}^\star := \sigma\big(\beta(Q_a^\star - Q_b^\star)\big), \qquad a, b \in \mathcal{A}.$$

First, we reformulate the loss of the theorem. For this purpose, we consider two cases:

1. When $a = b$: In this case the two logits coincide, so $P_{aa}^\star = P_{aa} = \sigma(0) = \frac{1}{2}$; hence in this case each summand equals $-\frac{1}{2} \log \frac{1}{2} = \frac{\log 2}{2}$. Summing over the $|\mathcal{A}|$ therefore contributes the constant $\frac{|\mathcal{A}| \log 2}{2 |\mathcal{A}|^2}$ in the loss.

2. For any two different actions $a \neq b$ the ordered pairs $(a, b)$ and $(b, a)$ both appear. Because $\sigma(z) + \sigma(-z) = 1$, we have the identities $P_{ba} = 1 - P_{ab}$ and $P_{ba}^\star = 1 - P_{ab}^\star$. Grouping those two ordered terms gives the compact expression

$$\mathcal{L}(\boldsymbol{\ell}) := -\frac{1}{|\mathcal{A}|^2} \sum_{\{a,b\} \in \mathcal{A}, a \neq b} \Big[ P_{ab}^\star \log P_{ab}(\boldsymbol{\ell}) + P_{ba}^\star \log P_{ba}(\boldsymbol{\ell}) \Big] \tag{10}$$

Therefore, $\mathcal{L}_{\text{pref}} = \frac{|\mathcal{A}| \log 2}{2 |\mathcal{A}|^2} + \mathcal{L}$. Since the additive constant is not used in the optimization, it can be discarded. Therefore, we may optimize $\mathcal{L}$ instead of $\mathcal{L}_{\text{pref}}$.

Now we characterize the stationary points. For this purpose, we compute the partial derivative of $\mathcal{L}$ with respect to the $\ell_k$, $\frac{\partial \mathcal{L}}{\partial \ell_k}$. Based on Lemma A.3 only the terms that contain $k$ depend on $\ell_k$, so

$$\frac{\partial \mathcal{L}}{\partial \ell_k} = -\frac{\alpha}{|\mathcal{A}|^2} \sum_{b \neq k} \left[ P_{kb}^\star - P_{kb}(\boldsymbol{\ell}) \right]. \tag{11}$$

A stationary point satisfies $\sum_{b \neq k}(P_{kb} - P_{kb}^\star) = 0$ for every $k$.

Set

$$\ell_a = c + \frac{\beta}{\alpha} Q_a^*, \qquad \forall a \in \mathcal{A} \tag{12}$$

Then for every $a, b$,

$$P_{ab}(\boldsymbol{\ell}) = \sigma\big(\alpha(\ell_a - \ell_b)\big) = \sigma\big(\beta(Q_a^* - Q_b^*)\big) = P_{ab}^*,$$

and equation 11 is equal to zero with $\ell_a = c + \frac{\beta}{\alpha} Q_a^*$.

Now using $\sum_a e^{\ell_a} = 1$ with equation 12 gives

$$e^c = \left( \sum_a \exp\big(\tfrac{\beta}{\alpha} Q_a^\star\big) \right)^{-1}.$$

Hence, the stationary point is as follows:

$$\pi_\star(a \mid s) = \frac{\exp(\tfrac{\beta}{\alpha} Q_a^\star)}{\displaystyle\sum_{a' \in \mathcal{A}} \exp(\tfrac{\beta}{\alpha} Q_{a'}^\star)}. \tag{13}$$

If we write the KKT conditions of the loss and derive the value of the Lagrange multiplier $\lambda$, $\lambda$ will be zero. Hence, the above stationary point is valid. See Lemma A.4 for the details.

**Computing Hessian:** To compute the Hessian we define $w_{ab}$ as follows:

$$w_{ab} := P_{ab}(\boldsymbol{\ell}) P_{ba}(\boldsymbol{\ell}) = P_{ab}(\boldsymbol{\ell})\big[1 - P_{ab}(\boldsymbol{\ell})\big] > 0.$$

Using

$$\frac{\partial P_{ab}}{\partial \ell_a} = +\alpha\, w_{ab}, \qquad \frac{\partial P_{ab}}{\partial \ell_b} = -\alpha\, w_{ab}.$$

Now, if we differentiate equation 11 once more. For $(i \neq j)$:

$$\frac{\partial^2 \mathcal{L}_{\text{p}}}{\partial \ell_j \, \partial \ell_i} = -\frac{\alpha}{|\mathcal{A}|^2} \big[\alpha\, w_{ij}\big] = -\frac{\alpha^2}{|\mathcal{A}|^2}\, w_{ij}.$$

For $(i = j)$:

$$\frac{\partial^2 \mathcal{L}_{\text{p}}}{\partial \ell_i^2} = -\frac{\alpha}{|\mathcal{A}|^2} \sum_{b \neq i} (-\alpha\, w_{ib}) = \frac{\alpha^2}{|\mathcal{A}|^2} \sum_{b \neq i} w_{ib}.$$

Now using the above derivations, we write the matrix form of Hessian.

$$\mathbf{H} = \frac{\alpha^2}{|\mathcal{A}|^2} (\mathbf{D} - \mathbf{W}) \qquad \begin{aligned} W_{ij} &= w_{ij}\ (i \neq j), \quad W_{ii} = 0, \\ D_{ii} &= \sum_{b \neq i} w_{ib}. \end{aligned}$$

To prove that $\mathcal{L}$ is strictly convex, we should prove that $\mathbf{H}$ (or $\mathbf{L}$) is positive-definite. The matrix $\mathbf{L} = \mathbf{D} - \mathbf{W}$ is a weighted graph Laplacian of the complete graph on $\mathcal{A}$ as its off–diagonal entries are negative, diagonals are positive, and each row sums to zero(Spielman, 2010).

For a matrix $\mathbf{L}$ to be positive definite, we should have for any $v \in \mathbb{R}^{|\mathcal{A}|}$, $v^\top \mathbf{L} v > 0$. In our case, one has

$$v^\top \mathbf{L} v = \frac{1}{2} \sum_{i,j} w_{ij}(v_i - v_j)^2. \tag{14}$$

Identity equation 14 follows from expanding $v^\top (\mathbf{D} - \mathbf{W})v$ and re-grouping terms (see Spielman (2010) for more details). Because every weight $w_{ij} > 0$, the RHS is non-negative, hence it is always equal to or bigger than zero. Therefore $\mathbf{L}$ is positive-semidifinite ($\mathbf{L} \succeq 0$) and equals to zero *iff*

$$v_1 = \cdots = v_{|\mathcal{A}|}.$$

In other words, only subspace $\text{span}\{\mathbf{1}\} = \{\, a\mathbf{1} : a \in \mathbb{R}, a \neq 0 \,\}$, whose members have all coordinates equal ($v_1 = \cdots = v_{|\mathcal{A}|}$) makes $v^\top \mathbf{L} v$ equal to zero. Now, we prove that given the constraint imposed on our problem, $v^\top \mathbf{L} v$ cannot be equal to zero.

In general unconstrained optimization, $v$ in $v^\top \mathbf{L} v$, shows all possible directions in $\mathbb{R}^{|\mathcal{A}|}$. In our case, the optimization is constrained, and the function $\mathcal{L}$ is restricted to a constraint set $C$ (the probability simplex: $\sum_{a \in \mathcal{A}} e^{\ell_a} - 1 = 0$), hence the condition $v^\top \mathbf{H} v \geq 0$ is only required for vectors $v$ in the tangent space of $C$ at $\ell$. This is because the tangent space of a convex set $C$ at any $v \in C$ is the set of feasible directions within $C$ (Absil et al., 2009).

The parameter set of the $\mathbf{H}$ (or accordingly $\mathbf{L}$) is

$$\mathcal{E} := \left\{ \boldsymbol{\ell} \in \mathbb{R}^{|\mathcal{A}|} \ : \ \sum_a e^{\ell_a} = 1, \ e^{\ell_a} > 0 \right\}$$

We define $g(\boldsymbol{\ell}) = \sum_{a \in \mathcal{A}} e^{\ell_a} - 1 = 0$ and its gradient $\nabla g(\boldsymbol{\ell}) = e^{\boldsymbol{\ell}} := (e^{\ell_1}, \dots, e^{\ell_{|\mathcal{A}|}})^\top > 0$. A displacement $v \in \mathbb{R}^{|\mathcal{A}|}$ is *feasible* iff it is in the tangent space (denote it with $T_{\boldsymbol{\ell}}\mathcal{E}$) that is $\nabla g(\boldsymbol{\ell}) \cdot v = 0$. Hence,

$$e^{\boldsymbol{\ell}} \cdot v \ = \ 0.$$

Now consider a vector in $\text{span}\{\mathbf{1}\}$. For any $v = a\mathbf{1}$ with $a \neq 0$,

$$e^{\boldsymbol{\ell}} \cdot v \ = \ a\, e^{\boldsymbol{\ell}} \cdot \mathbf{1} \ = \ a \sum_{b \in \mathcal{A}} e^{\ell_b} \ = \ a \ \neq \ 0.$$

Hence, $v \notin T_{\boldsymbol{\ell}}\mathcal{E}$. Hence, for every *feasible* $v \neq 0$,

$$v^\top \mathbf{L} v \ > \ 0 \quad \Longrightarrow \quad v^\top \mathbf{H} v = \frac{\alpha^2}{|\mathcal{A}|^2}\, v^\top \mathbf{L} v \ > \ 0.$$

Thus, the Hessian is positive–definite along all feasible directions, which establishes the strict convexity of the preference loss on the full support tabular policy.

**Another way to proof the uniqueness of the solution:**   Assume, for contradiction, that there exists another log–policy vector $\tilde{\boldsymbol{\ell}} \in \mathcal{E}$ that also satisfies the stationarity system $\sum_{b \neq k}(P_{kb} - P_{kb}^\star) = 0$, $\forall k$ and the normalization $\sum_a e^{\tilde{\ell}_a} = 1$. For every ordered pair $(k, j)$ the argument leading to equation **??** then gives $P_{kj}(\tilde{\boldsymbol{\ell}}) = P_{kj}^\star$ as well. Because $\sigma$ is strictly increasing, this implies

$$\tilde{\ell}_k - \tilde{\ell}_j \ = \ \ell_k - \ell_j, \qquad \forall k, j \in \mathcal{A},$$

hence $\tilde{\boldsymbol{\ell}} = \boldsymbol{\ell} + \delta\mathbf{1}$ for some $\delta \in \mathbb{R} \setminus \{0\}$. But then

$$\sum_a e^{\tilde{\ell}_a} \ = \ e^\delta \sum_a e^{\ell_a} \ = \ e^\delta \neq 1,$$

contradicting the constraint that every feasible $\boldsymbol{\ell}$ must satisfy $\sum_a e^{\ell_a} = 1$. Therefore $\delta = 0$ and $\tilde{\boldsymbol{\ell}} = \boldsymbol{\ell}$, proving that the stationary point is unique.

**Connection with soft actor-critic:** For the same fixed state consider

$$J_\tau(\pi) := \mathbb{E}_{a \sim \pi}[Q_a^\star] + \tau \mathcal{H}(\pi), \qquad \mathcal{H}(\pi) := -\sum_a \pi(a) \log \pi(a).$$

Introducing a Lagrange multiplier $\lambda$ for $\sum_a \pi(a) = 1$ gives $Q_a^\star - \tau(\log \pi(a) + 1) - \lambda = 0$, hence $\pi(a) \propto \exp(Q_a^\star/\tau)$. Normalization produces

$$\pi_{\mathrm{SAC}}(a \mid s) = \frac{\exp(Q_a^\star/\tau)}{\sum_{a'} \exp(Q_{a'}^\star/\tau)}.$$

Choosing $\tau = \alpha/\beta$ recovers equation 13, so the minimizer of $\mathcal{L}$ coincides with the soft actor-critic solution with temperature $\tau = \alpha/\beta$.

$\square$

**Lemma A.3** (Gradient of the unordered-pair preference loss). *Let*

$$\mathcal{L}(\boldsymbol{\ell}) := -\frac{1}{|\mathcal{A}|^2} \sum_{\{a,b\} \in \mathcal{A}, a \neq b} \Big[ P_{ab}^\star \log P_{ab}(\boldsymbol{\ell}) + P_{ba}^\star \log P_{ba}(\boldsymbol{\ell}) \Big],$$

*where $P_{ab}^\star$, $P_{ab}(\boldsymbol{\ell})$ and $\boldsymbol{\ell}$ are defined in Theorem A.2. Then, for every action $k \in \mathcal{A}$,*

$$\frac{\partial \mathcal{L}}{\partial \ell_k} = -\frac{\alpha}{|\mathcal{A}|^2} \sum_{b \neq k} \big[ P_{kb}^\star - P_{kb}(\boldsymbol{\ell}) \big]. \tag{15}$$

*Proof.* For each unordered pair $\{a, b\}$, define

$$g_{ab}(\boldsymbol{\ell}) := P_{ab}^\star \log P_{ab}(\boldsymbol{\ell}) + P_{ba}^\star \log P_{ba}(\boldsymbol{\ell}).$$

Because $\dfrac{d}{dz} \log \sigma(z) = 1 - \sigma(z)$ and $\partial(\ell_a - \ell_b)/\partial \ell_k = \mathbf{1}\{k = a\} - \mathbf{1}\{k = b\}$,

$$\frac{\partial}{\partial \ell_k} \log P_{ab} = \alpha(1 - P_{ab}) \big[ \mathbf{1}\{k = a\} - \mathbf{1}\{k = b\} \big],$$

$$\frac{\partial}{\partial \ell_k} \log P_{ba} = \alpha P_{ab} \big[ \mathbf{1}\{k = b\} - \mathbf{1}\{k = a\} \big].$$

Since $P_{ba}^\star = 1 - P_{ab}^\star$ and $P_{ba} = 1 - P_{ab}$,

$$\frac{\partial g_{ab}}{\partial \ell_k} = \alpha(\mathbf{1}\{k = a\} - \mathbf{1}\{k = b\}) \big[ P_{ab}^\star - P_{ab} \big]. \tag{16}$$

Insert equation 16 into the loss and sum over all unordered pairs that contain $k$:

$$\frac{\partial \mathcal{L}}{\partial \ell_k} = -\frac{\alpha}{|\mathcal{A}|^2} \sum_{\{a,b\}} (\mathbf{1}\{k = a\} - \mathbf{1}\{k = b\}) \big[ P_{ab}^\star - P_{ab} \big].$$

If $k = a$ (and $b > k$) the indicator equals $+1$; if $k = b$ (with $a < k$) it equals $-1$. Using again the symmetry $P_{ak}^\star = 1 - P_{ka}^\star$ and $P_{ak} = 1 - P_{ka}$, the negative sign flips the difference so that both cases contribute the same quantity $P_{kb}^\star - P_{kb}$. Hence

$$\frac{\partial \mathcal{L}}{\partial \ell_k} = -\frac{\alpha}{|\mathcal{A}|^2} \sum_{b \neq k} \big[ P_{kb}^\star - P_{kb}(\boldsymbol{\ell}) \big],$$

completing the proof. $\square$

**Lemma A.4** (The KKT multiplier). *Consider the constrained minimization of the unordered-pair preference loss*

$$\min_{\boldsymbol{\ell} \in \mathbb{R}^{|\mathcal{A}|}} \mathcal{L}(\boldsymbol{\ell}) \quad s.t. \quad g(\boldsymbol{\ell}) := \sum_{a \in \mathcal{A}} e^{\ell_a} - 1 = 0,$$

*with $\mathcal{L}$ defined in equation 10. Let $\lambda \in \mathbb{R}$ be the Lagrange multiplier associated with the normalization constraint. At every KKT point $(\boldsymbol{\ell}, \lambda)$ one necessarily has*

$$\lambda = 0.$$

*Proof.* The KKT stationarity condition for each action $k \in \mathcal{A}$ is

$$-\frac{\alpha}{|\mathcal{A}|^2} \sum_{b \neq k} \left[ P_{kb}^\star - P_{kb}(\boldsymbol{\ell}) \right] + \lambda\, e^{\ell_k} = 0. \tag{16}$$

Summing equation 16 over all $k$ gives

$$-\frac{\alpha}{|\mathcal{A}|^2} \sum_{k} \sum_{b \neq k} \left[ P_{kb}^\star - P_{kb}(\boldsymbol{\ell}) \right] + \lambda \sum_{k} e^{\ell_k} = 0. \tag{17}$$

Because the log–policy variables satisfy the equality constraint $\sum_k e^{\ell_k} = 1$, the second term in equation 17 sums to $\lambda$.

Rewrite the double sum by grouping every *ordered* pair $(k, b)$ with its reverse $(b, k)$:

$$\sum_{k} \sum_{b \neq k} \left[ P_{kb}^\star - P_{kb} \right] = \sum_{\substack{k, b \in \mathcal{A} \\ k < b}} \left[ (P_{kb}^\star - P_{kb}) + (P_{bk}^\star - P_{bk}) \right].$$

Using the fact that $P_{kb}^\star + P_{bk}^\star = 1$ and $P_{kb} + P_{bk} = 1$, each term in the bracket equals to $1 - 1 = 0$. Therefore, the entire double sum is zero, and we can imply that $\lambda = 0$.

$\square$

# B   TRAJECTORY–LEVEL ANALYSIS OF DFA

Let $\mathcal{T}_H$ be the (finite) set of all length–$H$ trajectories $\tau = (s_0, a_0, \ldots, s_{H-1}, a_{H-1})$ that can be generated by the MDP.

**Assumption B.1** (Trajectory-level Bradley–Terry model). Let

$$G^\star(\tau) := \sum_{t=0}^{H-1} \gamma^t\, r(s_t, a_t)$$

be the *return* of trajectory $\tau$. There exists $\beta > 0$ such that for every pair $\tau_1, \tau_2 \in \mathcal{T}_H$

$$P^\star(\tau_1 \succ \tau_2) = \sigma\big(\beta\, [G^\star(\tau_1) - G^\star(\tau_2)]\big), \qquad \sigma(z) = \frac{1}{1 + e^{-z}}.$$

## B.1   TRAJECTORY PREFERENCE LOSS

Parameterise a *trajectory-tabular* policy by one log-likelihood per path, $L_\tau = \log \pi_\theta(\tau)$, subject to the simplex constraint $\sum_{\tau \in \mathcal{T}_H} e^{L_\tau} = 1$, $e^{L_\tau} > 0$. For ordered trajectory pairs sampled uniformly from $\mathcal{T}_H^2$ define the loss

$$\mathcal{L}_{\text{traj}}(L) := -\frac{1}{|\mathcal{T}_H|^2} \sum_{\tau_1, \tau_2 \in \mathcal{T}_H} P^\star(\tau_1 \succ \tau_2) \log \sigma\big(\alpha[L_{\tau_1} - L_{\tau_2}]\big), \tag{18}$$

with parameter $\alpha > 0$.

**Theorem B.2** (Optimal policy for trajectory loss). *Assume Theorem B.1 and uniform sampling of ordered trajectory pairs. The loss equation 18 is strictly convex on the probability simplex $\{L : \sum_\tau e^{L_\tau} = 1\}$ and attains its* unique *minimum at the Gibbs distribution*

$$\pi_\star(\tau) = \frac{\exp\big(\frac{\beta}{\alpha}\, G^\star(\tau)\big)}{\sum_{\tau' \in \mathcal{T}_H} \exp\big(\frac{\beta}{\alpha}\, G^\star(\tau')\big)}. \tag{19}$$

*The proof is similar to the proof of Theorem 5.2*

## B.2 Connection between the State-wise and Trajectory-wise Optima

We adopt the standard soft Bellman optimality equations (See Haarnoja et al. (2017) for the proofs):

$$Q^\star(s,a) = r(s,a) + \gamma\,\mathbb{E}_{s'\sim P(\cdot|s,a)}\big[V^\star(s')\big], \tag{20}$$

$$V^\star(s) = \alpha/\beta \log \sum_{a'\in\mathcal{A}} \exp\!\big(\tfrac{\beta}{\alpha}Q^\star(s,a')\big). \tag{21}$$

Let $\alpha, \beta > 0$ be the preference and BT-scale parameters. The optimal (state-wise) policy can be written equivalently as

$$\pi_{\mathrm{st}}(a\mid s) = \frac{\exp\!\big(\tfrac{\beta}{\alpha}Q^\star(s,a)\big)}{\sum_{a'}\exp\!\big(\tfrac{\beta}{\alpha}Q^\star(s,a')\big)} = \exp\!\Big(\tfrac{\beta}{\alpha}\big(Q^\star(s,a) - V^\star(s)\big)\Big). \tag{22}$$

Consider a finite-horizon trajectory $\tau = (s_0, a_0, \ldots, s_{H-1}, a_{H-1})$ generated by the deterministic dynamics $s_{t+1} = f(s_t, a_t)$, with fixed initial state $s_0$ and terminal boundary condition $V^\star(s_H) = 0$. Define the trajectory probability induced by the state-wise policy,

$$\pi_{\mathrm{traj}}(\tau) := \prod_{t=0}^{H-1} \pi_{\mathrm{st}}(a_t\mid s_t). \tag{23}$$

Taking logs and using equation 22,

$$\log \pi_{\mathrm{traj}}(\tau) = \frac{\beta}{\alpha}\sum_{t=0}^{H-1}\big(Q^\star(s_t, a_t) - V^\star(s_t)\big). \tag{24}$$

Under the assumptions $\gamma = 1$ and deterministic transitions, the soft Bellman equation equation 20 reduces to

$$Q^\star(s_t, a_t) = r_t + V^\star(s_{t+1}), \tag{25}$$

hence

$$Q^\star(s_t, a_t) - V^\star(s_t) = r_t + V^\star(s_{t+1}) - V^\star(s_t). \tag{26}$$

Summing over $t = 0, \ldots, H-1$ telescopes the value terms:

$$\sum_{t=0}^{H-1}\big(Q^\star(s_t, a_t) - V^\star(s_t)\big) = \sum_{t=0}^{H-1} r_t + V^\star(s_H) - V^\star(s_0) = \sum_{t=0}^{H-1} r_t - V^\star(s_0). \tag{27}$$

Combining equation 24 and equation 27 gives

$$\log \pi_{\mathrm{traj}}(\tau) = \frac{\beta}{\alpha}\Big(\sum_{t=0}^{H-1} r_t - V^\star(s_0)\Big). \tag{28}$$

Therefore, under these conditions, for any trajectory $\tau$ starting from a fixed $s_0$ we have

$$\pi_{\mathrm{traj}}(\tau) = \exp\!\Big(\tfrac{\beta}{\alpha}\big(G(\tau) - V^\star(s_0)\big)\Big) = C(s_0)\exp\!\Big(\tfrac{\beta}{\alpha}G(\tau)\Big) \propto \exp\!\Big(\tfrac{\beta}{\alpha}G(\tau)\Big),$$

where $G(\tau) = \sum_{t=0}^{H-1} r_t$ and $C(s_0) = \exp\!\big(-\tfrac{\beta}{\alpha}V^\star(s_0)\big)$ depends only on the common start state.

This establishes that the trajectory probabilities induced by the state-wise soft-optimal policy are proportional to $\exp\big((\beta/\alpha)G(\tau)\big)$ (for fixed $s_0$), completing the connection between the state-wise and trajectory-wise formulations.

Moreover, for any two trajectories $\tau^+, \tau^-$ with the same $s_0$,

$$\alpha\big[\log \pi_{\mathrm{traj}}(\tau^+) - \log \pi_{\mathrm{traj}}(\tau^-)\big] = \beta\big[G(\tau^+) - G(\tau^-)\big],$$

so the DFA trajectory-level preference computed from $\pi_{\mathrm{traj}}$ reduces exactly to the Bradley–Terry model over undiscounted returns:

$$P_\theta(\tau^+ \succ \tau^-) = \sigma\Big(\beta\,[G(\tau^+) - G(\tau^-)]\Big).$$

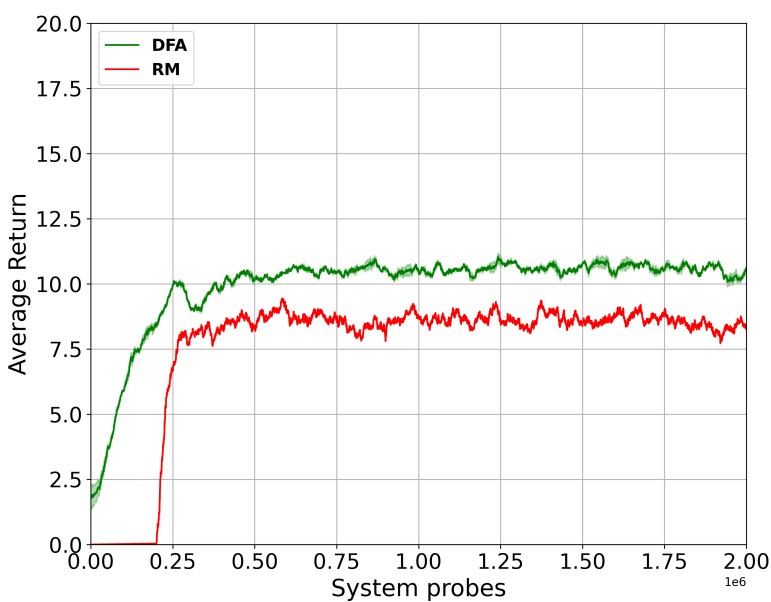

Figure 4: GridWorld results with size 10*10. ($M = 500$)

## C    EXPERIMENTS

Walker is a planar biped with four actuated joints that must walk without tipping over; Hopper is a one-leg, three-joint robot that learns to hop forward; Humanoid is a 17-joint 3D figure that must walk quickly while remaining upright; Swimmer is a three-link snake that propels itself through a viscous medium; Inverted Pendulum tasks a cart with balancing an upright pole; and MountainCar Continuous challenges a car trapped between two hills to climb the right hill by building momentum.

For GridWorld game, the grid is 5*5, where the agent starts at position 2*2. All methods use a tabular softmax policy parameterization, where each state-action pair has a corresponding logit parameter. For the reward model in RM+PPO, we use a simple tabular representation that assigns a value to each state-action pair. All methods are trained for the same number of environment interactions to ensure fair comparison. We use Adam optimizer with learning rate $3 \times 10^{-2}$ across all methods.

Below we illustrate the results for more complex environments and different numbers of $M$ mentioned in Section 6. We run the algorithms for 5 different seeds $\{3, 1, 14, 4, 50\}$. We considered the default gym horizon for all environments.

Figure 7 shows the performance of DFA on Humanoid environments compared with OraclePPO and RM methods. For the panel query, with used M=10. For the RM method, we spent 2000 system probes for the reward model training. In tasks whose rewards are easy to model, the performance gap between RM and Oracle PPO is typically small; we therefore chose Humanoid, where the reward is harder to model than in the other environments.

We utilized a Linux server with Intel Xeon CPU E5-2680 v3 (24 cores) operating at 2.50GHz with 377 GB DDR4 of memory and Nvidia Titan X Pascal GPU. The computation was distributed over 48 threads to ensure a relatively efficient run time. In our control-task experiments, DFA required more wall-clock time than SAC. For example, in the Pendulum, running 10 million system probes took 8 hours (on average) with DFA compared to 6.5 hours with SAC. In the Swimmer environment, SAC completed in 8 hours, while DFA took 9 hours. Although DFA generally requires more wall-clock time per step, in some environments (e.g., MountainCar, Swimmer) it converges in fewer steps. As a result, the increased per-step runtime does not significantly impact its overall efficiency.

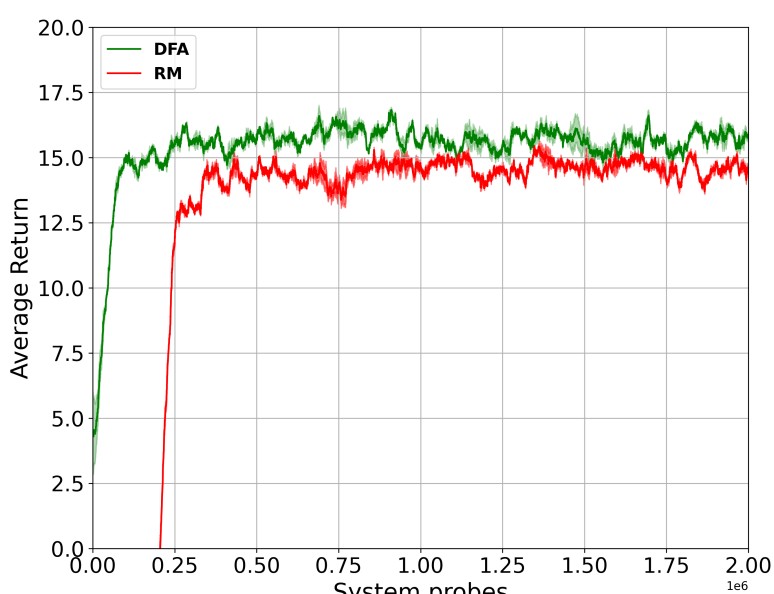

Figure 5: GridWorld results with size 20*20($M = 100$).

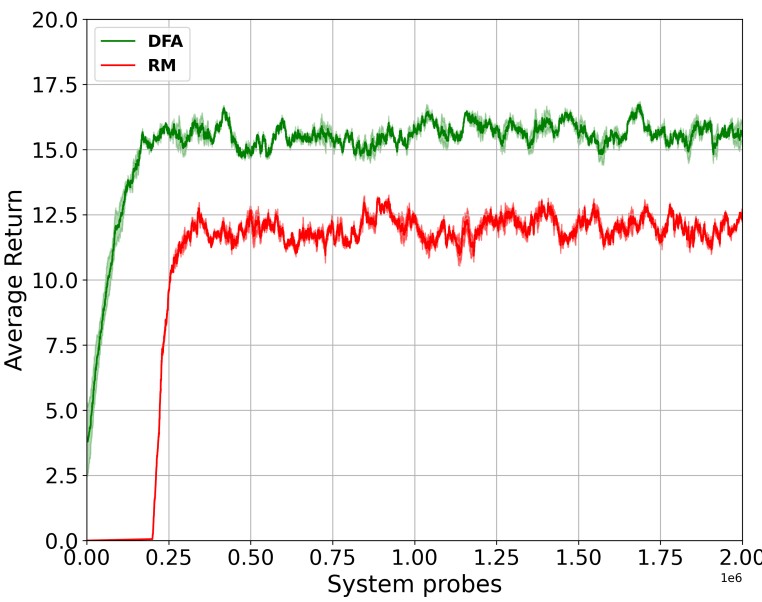

Figure 6: GridWorld results with size $20 \times 20(M = 1)$.

For hyperparameter tuning, we performed a grid search, systematically exploring a predefined range of values for each parameter. In the following tables, we provide the fine-tuned parameters for each algorithm and method. Batch sizes are considered the same for all algorithms. The discount factor is also set to 0.99 for all the runs.

Table 1: Default hyper-parameters for all algorithms used in the $5 \times 5$ GridWorld experiments

| Algorithm | Hyper-parameter | Value |
|---|---|---|
| ZPG | $T$ (iterations) | 1000000 |
| | $N$ (pairs / iter) | 1 - 10 |
| | $M$ (votes / query) | 1000 |
| | $\mu$ (perturbation radius) | 0.1 |
| | $\alpha$ (learning-rate) | 0.05 |
| | trim (prob. clip) | $10^{-2}$ |
| RM-PPO | traj_pairs (pretaining) | 5000 |
| | ppo_iters | 1000 |
| | $\beta_{\mathrm{KL}}$ | 0.1 |
| | $\gamma$ (discount) | 1.0 |
| | $\lambda$ (GAE) | 0.95 |
| DFA (on-policy) | $\alpha$ (temperature) | $1 \times 10^{-3} - 1 \times 10^{-6}$ |
| | $N_{\text{pairs/iter}}$ | 1 |
| | iters | 100000 |
| Oracle-PPO | ppo_iters | 100000 |
| | $\beta_{\mathrm{KL}}$ | 0.1 |
| | $\gamma$ (discount) | 1.0 |
| | $\lambda$ (GAE) | 0.95 |

Table 2: Hyper-parameters for SAC and DFA across all evaluated environments

| Alg. | Hyper-parameter | Walker2d | Hopper | Swimmer | Humanoid | MountainCarC | Pendulum |
|---|---|---|---|---|---|---|---|
| SAC | Hidden layer size | 64 | 64 | 64 | 64 | 64 | 64 |
| | Policy learning-rate | $1 \times 10^{-3}$ | $1 \times 10^{-3}$ | $1 \times 10^{-3}$ | $1 \times 10^{-3}$ | $1 \times 10^{-3}$ | $1 \times 10^{-3}$ |
| | $Q$ learning-rate | $1 \times 10^{-3}$ | $1 \times 10^{-3}$ | $1 \times 10^{-3}$ | $1 \times 10^{-3}$ | $1 \times 10^{-3}$ | $1 \times 10^{-3}$ |
| | Batch size | 256 | 256 | 256 | 256 | 256 | 256 |
| | Replay-buffer capacity | 20 000 | 20 000 | 20 000 | 20 000 | 20 000 | 20 000 |
| | Entropy temperature $\lambda$ | 0.1 | 0.2 | 0.01 | 0.01 | 0.1 | 0.2 |
| | Discount factor $\gamma$ | 0.99 | 0.99 | 0.99 | 0.99 | 0.99 | 0.99 |
| | Soft-update coefficient $\tau$ | 0.1 | 0.005 | 0.1 | 0.1 | 0.01 | 0.005 |
| | # parallel envs $N_{\text{env}}$ | 32 | 32 | 32 | 32 | 32 | 32 |
| | Training episodes | 50 000 | 50 000 | 50 000 | 50 000 | 50 000 | 50 000 |
| DFA | Hidden layer size | 64 | 64 | 64 | 64 | 64 | 64 |
| | Policy learning-rate | $1 \times 10^{-3}$ | $1 \times 10^{-3}$ | $1 \times 10^{-3}$ | $1 \times 10^{-3}$ | $1 \times 10^{-3}$ | $1 \times 10^{-3}$ |
| | $Q$ learning-rate | $1 \times 10^{-3}$ | $1 \times 10^{-3}$ | $1 \times 10^{-3}$ | $1 \times 10^{-3}$ | $1 \times 10^{-3}$ | $1 \times 10^{-3}$ |
| | Batch size | 256 | 256 | 256 | 256 | 256 | 256 |
| | Replay-buffer capacity | 20 000 | 20 000 | 20 000 | 20 000 | 20 000 | 20 000 |
| | Entropy temperature $\lambda$ | 0.01 | 0.1 | 0.01 | 0.01 | 0.01 | 0.01 |
| | Temperature $\alpha$ | 0.2 | 0.2 | 0.3 | 0.2 | 0.4 | 0.2 |
| | Discount factor $\gamma$ | 0.99 | 0.99 | 0.99 | 0.99 | 0.99 | 0.99 |
| | Soft-update coefficient $\tau$ | 0.1 | 0.005 | 0.1 | 0.1 | 0.01 | 0.005 |
| | # parallel envs $N_{\text{env}}$ | 32 | 32 | 32 | 32 | 32 | 32 |
| | Training episodes | 50 000 | 50 000 | 50 000 | 50 000 | 50 000 | 50 000 |

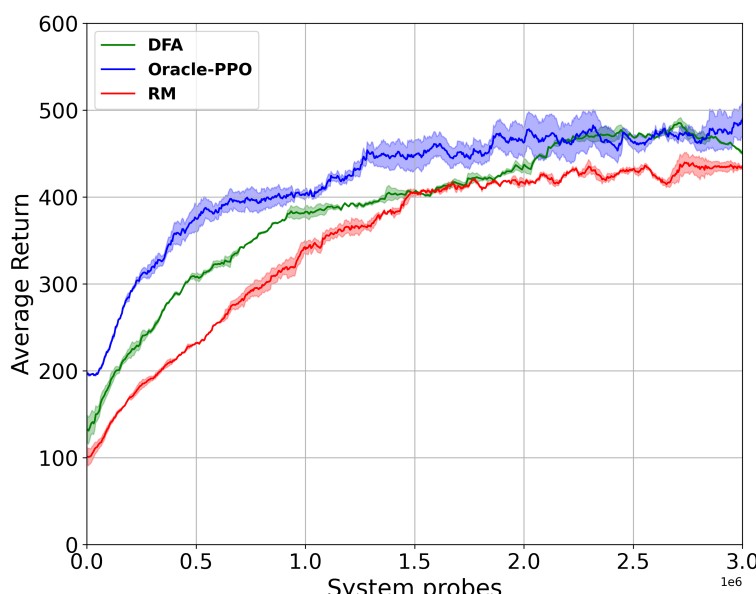

Figure 7: Humanoid results with horizon 1000.

## D  BROADER IMPACTS

DFA aims to make reinforcement learning from human feedback more sample-efficient by blending numeric rewards with pairwise preferences. Positive impacts include lowering annotation costs, enabling faster prototyping of assistive robots, and providing a simple baseline for preference-centric alignment research. However, the method also amplifies whatever biases or inconsistencies are present in the collected preferences: if early $Q$ estimates or human labels encode unfair or unsafe behavior, DFA may reinforce those patterns more quickly than reward-only training. Because DFA can learn from very small amounts of feedback, malicious or accidental injection of adversarial comparisons could steer policies toward harmful objectives—especially in safety-critical domains such as autonomous driving or content recommendation. The work uses only simulated environments and involves no personal data; nevertheless, broader deployment should respect fairness guidelines and, when real users provide feedback, comply with relevant privacy regulations.

## E  USE OF LARGE LANGUAGE MODELS

We used Large Language Models (LLMs) to aid or polish the manuscript text. Specifically, LLMs were used to improve grammar, phrasing, and clarity of exposition; they were also used for code debugging.

## F  REBUTTAL EXPERIMENT

We added one more experiment during the rebuttal discussion on a new environment in MetaWorld, requested by the reviewers. The experiment setup is consistent with the paper Liu et al. (2022). We will update this section for the camera-ready version with more experiments.

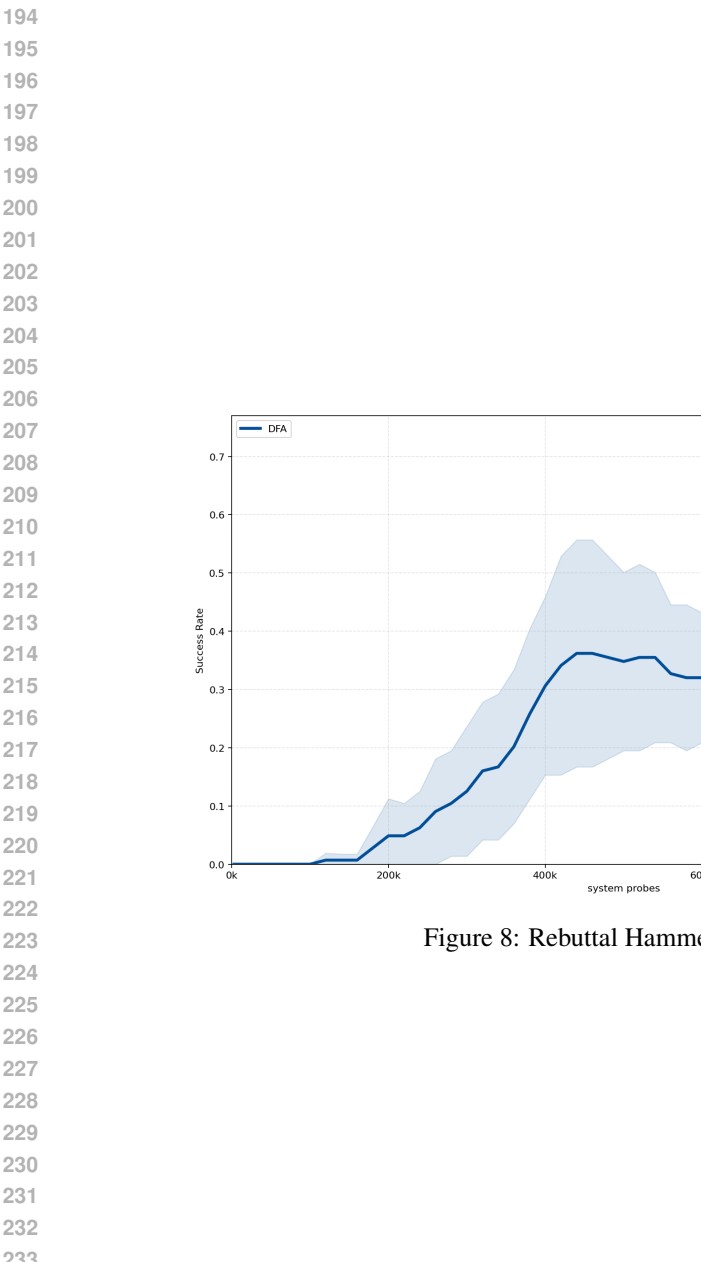

Figure 8: Rebuttal Hammer results.

