# OpenReview forum: "Fusing Rewards and Preferences in Reinforcement Learning"
_ICLR.cc/2026/Conference — Submitted to ICLR 2026_

### Official Review · Reviewer_2Mgv · 2025-10-28

**Soundness:** 3
**Presentation:** 3
**Contribution:** 3
**Rating:** 4
**Confidence:** 3

**Summary:**

This paper proposes the Dual-Feedback Actor (DFA) algorithm for RL, which fuses both scalar reward signals and pairwise preferences into a unified training objective. DFA uses the agent’s policy log-probabilities to model preferences, bypassing explicit reward modeling. The authors prove that under the Bradley-Terry model, minimizing the preference loss recovers the Soft Actor-Critic (SAC) policy. Empirically, DFA matches or exceeds SAC in control tasks and outperforms preference-based baselines like RM+PPO and ZPG in stochastic settings.

**Strengths:**

1. The paper is well-written and easy to follow, providing enough background with appropriate notations.

2. The proposed algorithm is well-motivated and theoretically grounded. The method is simple yet interesting.

**Weaknesses:**

My main concerns lie with the experimental evaluation, which I believe is currently insufficient to fully support the paper’s claims. Specifically:

1. The paper emphasizes the ability of DFA to handle both numerical rewards and preference feedback, but it is unclear whether the method remains effective when both signals are present simultaneously. It would be valuable to include an experiment that directly evaluates this mixed-signal scenario.

2. In Section 6.1, comparing DFA only with SAC is insufficient and "SAC the strongest baseline on many environments" seems not correct. The authors should include additional baselines such as PPO, TD3 [1], and Rainbow [2] to establish a stronger empirical foundation.

3. In Section 6.2, more preference-based baselines should be considered, especially DPO and IPL [3], which have become standard in RLHF research. Furthermore, experiments should extend beyond the synthetic GridWorld environment. Consider using more challenging and realistic continuous domains that incorporate stochasticity, such as risk-sensitive D4RL[4].

4. Finally, the current experimental setup assumes stochasticity only in the environment’s transitions. A more realistic evaluation should consider noisy or inconsistent preferences, or noisy rewads like [4], which are common in real-world RLHF scenarios. Can DFA handle such noisy human feedback?

Overall, the proposed method is theoretically promising and well-positioned within the literature. However, to fully demonstrate its practical value, the experimental section needs to be significantly expanded. I would be willing to consider increasing my score if the authors address these concerns in the rebuttal.

[1] Hessel M, Modayil J, Van Hasselt H, et al. Rainbow: Combining improvements in deep reinforcement learning[C]//Proceedings of the AAAI conference on artificial intelligence. 2018, 32(1).

[2] Fujimoto S, Hoof H, Meger D. Addressing function approximation error in actor-critic methods[C]//International conference on machine learning. PMLR, 2018: 1587-1596.

[3] Hejna J, Sadigh D. Inverse preference learning: Preference-based rl without a reward function[J]. Advances in Neural Information Processing Systems, 2023, 36: 18806-18827.

[4] Urpí N A, Curi S, Krause A. Risk-averse offline reinforcement learning[J]. arXiv preprint arXiv:2102.05371, 2021.

**Questions:**

see above

---

> ### Author Response · Authors · 2025-11-23
>
> We thank the reviewer for their valuable comments and address them below:
>
> >*The paper emphasizes the ability of DFA to handle both numerical rewards and preference feedback...*
>
> In this work, we followed the standard RLHF evaluation protocol for continuous control by simulating preferences (via a Bradley-Terry model with controlled noise), which is common practice for ensuring reproducibility. Preferences are simulated using the rewards.
> We agree that large-scale real human-in-the-loop studies are valuable future work. However, running such studies with substantial subject pools requires dedicated budget, logistics, and (often) IRB/ethics approvals, which are beyond the scope of the present submission. Importantly, our theoretical guarantees are agnostic to whether preferences are human- or synthetically generated. The results hinge on the preference model, not on the data collection modality. Methodologically, DFA natively supports both state-wise and trajectory-wise human labels without any modification to the loss or training loop.
>
> -------------
>
> >*In Section 6.1, comparing DFA only with SAC is insufficient and "SAC the strongest baseline ...*
>
> As stated in the paper, we initially focused on SAC because (i) extensive public benchmarks (e.g., \href{https://spinningup.openai.com/en/latest/spinningup/bench.html#benchmarks-for-spinning-up-implementations}{OpenAI SpinningUp}) consistently find SAC to be a top off-policy baseline on these continuous-control tasks (often matching or outperforming TD3 under comparable budgets), and (ii) our theory explicitly recovers the SAC policy under the Bradley--Terry preference model, making SAC the most directly relevant and strongest reward-based point of comparison.
>     Our submission already evaluates DFA on six standard MuJoCo control tasks (Walker2d, Hopper, Swimmer, Humanoid, InvertedPendulum, and MountainCarContinuous) and on multiple stochastic GridWorld settings that vary grid size, annotator pool size, and action-flip dynamics; more challenging GridWorld variants and complex MuJoCo environment (Humanoid) under preference supervision were provided in the appendix. To further extend the empirical scope beyond locomotion and grid settings, we are currently running additional experiments on manipulation tasks from the Meta-World benchmark under both reward- and preference supervision. We will include these results as soon as they are available.
>
> -------------
>
> > *In Section 6.2, more preference-based baselines should be considered...*
>
> We appreciate the suggestion and note that our current paper already includes DPO (we use the online variant), reward-modeling pipelines (RM+PPO), and ZPG in Sec. 6.2, with more challenging settings provided in the appendix: larger GridWorld variants (e.g., $10\cdot10$, $20\cdot20$ and with varying panel sizes $M$) and a complex continuous-control task (Humanoid). To further strengthen Section~6.2 in the revised version, we will add IPL baselines on more complex domains.
>
> ---------
>
> >*Finally, the current experimental setup assumes stochasticity only in the environment’s transitions...*
>
> Empirically, our experiments already operate with a non-converged Q-function, which is far from optimal for much of training (i.e., it is a noisy $Q_k$), yet DFA trains stably and often more smoothly than SAC and reward-modeling baselines. This suggests the method is robust in practice to imperfect critics.
>
> --------------
> Thank you once again for your feedback. We hope our responses have addressed your concerns and sincerely appreciate your consideration. If there are any additional questions or points that require clarification, please do not hesitate to let us know.

---

> ### Author Response · Authors · 2025-12-03
>
> Dear Reviewer,
>
> As promised, aside from the previous experiments, we conducted more experiments on the Meta-World Hammer task in a new environment, using settings similar to those in the referenced paper [5]. Compared to the results reported there, our method outperforms other PbRL approaches and achieves performance very close to SAC. Moreover, our experiment is more challenging because the teacher script preference data we generate is noisy (sampled from the BT model), whereas the other paper uses deterministic preference data.
>
> We have included our experiment in the last section of the paper PDF for your reference (Appendix F), and we plan to add more experiments before the camera-ready version. Due to time constraints, we were only able to include this experiment for now.
>
> [5] Meta-Reward-Net: Implicitly Differentiable Reward Learning for Preference-based Reinforcement Learning. 2022.
>
>
> Thank you once again for your feedback. We hope our responses have addressed your concerns and sincerely appreciate your consideration.

---

### Official Review · Reviewer_Gbv4 · 2025-10-31

**Soundness:** 2
**Presentation:** 2
**Contribution:** 2
**Rating:** 2
**Confidence:** 4

**Summary:**

This paper introduces an RL method that combines two types of feedback: scalar rewards and human preferences to enhance prefromance.

**Strengths:**

This paper claims dual compatibility with both reward signals and preference feedback, and shows that the method can be used in both on-policy and off-policy settings.

**Weaknesses:**

The motivation for using dual feedback is not clearly explained. In addition, the experiments are not solid. The paper lacks strong baselines, it should compare against standard PbRL (preference-based RL) methods as well as common rl algorithms to properly demonstrate effectiveness. More diverse environments are also needed to support the claims.

**Questions:**

I hope authors can clarify the motivation for using dual feedback. If scalar rewards are already available, why are human preferences still needed? If the goal is to show that DFA improves performance on difficult tasks, then the paper should include experiments on challenging environments. Currently, no such tasks are evaluated, which makes the claim insufficiently supported.

The paper spends substantial space discussing RLHF and LLM algorithm. Since this work focuses on PbRL for control tasks, I believe it would be more appropriate to review PbRL literature in robotics and control, rather than LLM-centric work.

For synthesizing preferences, are Q-values computed using the ground-truth reward, or with the reward learned from preferences? This detail is important for understanding the learning pipeline.

How many preference labels are used for each task?

I also suggest including standard environments commonly used in PbRL research, such as MetaWorld and dm_control. In addition, more baselines should be included, such as classical PbRL methods like PEBBLE [1] and MRN[2], to provide a stronger and fairer comparison.

[1] PEBBLE: Feedback-Efficient Interactive Reinforcement Learning via Relabeling Experience and Unsupervised Pre-training. 2021.\
[2] Meta-Reward-Net: Implicitly Differentiable Reward Learning for Preference-based Reinforcement Learning. 2022.

---

> ### Author Response · Authors · 2025-11-23
>
> We thank the reviewer for their valuable comments and address them below:
>
> >*The motivation for using dual feedback is not clearly explained.*
>
> We thank the reviewer for raising the question about motivation. Our motivation for \emph{dual} feedback is to unify two practically common supervision modes in RL—scalar rewards (when available, possibly sparse/noisy) and pairwise preferences (human or synthetic), into a single principled policy update that preserves entropy-regularized exploration. For example, the state-wise feedback is practical in many applications: e.g., when a mobile robot is stuck at a corner or blocked by an obstacle, a human can quickly indicate a preferred direction (turn left vs. reverse) as a pairwise action choice; interactive navigation, teleoperation, and TAMER-style coaching provide state-level feedbacks [3][4]. In many real settings, numeric rewards may be partially specified or costly to shape, while preference labels are easier to elicit but often limited; DFA is designed to seamlessly exploit either source, or both, without an explicit reward-modeling step. Even in our experiments, with binary feedback, we are outperforming SAC that has access to reward but uses noisy Q. One explanation could be that with synthesizing preference, we are denoising Q!
>
> >[1]: TAMER: Training an Agent Manually via Evaluative Reinforcement:Knox, W. Bradley and Stone, Peter
>
> >[2] Deep TAMER: Interactive Agent Shaping in High-Dimensional State Spaces: Warnell, Garrett and Waytowich, Nicholas and Lawhern, Vernon and Stone, Peter.
> ------------
>
> > *For synthesizing preferences, are Q-values computed using ...*
>
> For synthesizing preferences, the Q function is trained exactly as in SAC on the environment reward, and the resulting $Q$-values are used \emph{only} to synthesize preference pairs online. No reward model learned from preferences is used in this setting.
>      For MuJoCo, synthesized preferences are generated at each update step, where we draw a minibatch of $N{=}256$ states from the replay buffer and form one pair per state. Thus, the total number of \emph{synthetic} pairs equals $256\times$ (number of gradient updates). Because these are not human queries, we report results versus environment steps (system probes).
>
> ----------
>
> > *The paper spends substantial space discussing RLHF and LLM algorithm...The paper lacks strong baselines, it should compare against standard PbRL ...*
>
> We appreciate the suggestion to broaden the experimental evaluation. Our submission already evaluates DFA on six standard MuJoCo control tasks (Walker2d, Hopper, Swimmer, Humanoid, InvertedPendulum, and MountainCarContinuous) and on multiple stochastic GridWorld settings that vary grid size, annotator pool size, and action-flip dynamics; more challenging GridWorld variants and complex MuJoCo environment (Humanoid) under preference supervision were provided in the appendix C. To further extend the empirical scope beyond locomotion and grid settings, we are currently running additional experiments on manipulation tasks from the Meta-World benchmark, comparing with methods you mentioned (PEBBLE, MRN); we will include these results as soon as they are available.
>
>
> ------------------
>
>
> Thank you once again for your feedback. We hope our responses have addressed your concerns and sincerely appreciate your consideration. If there are any additional questions or points that require clarification, please do not hesitate to let us know.

---

> ### Author Response · Authors · 2025-12-03
>
> Dear Reviewer,
>
> As promised, aside from the previous experiments, we conducted more experiments on the Meta-World Hammer task in a new environment, using settings similar to those in the referenced paper [5]. Compared to the results reported there, our method outperforms other PbRL approaches and achieves performance very close to SAC. Moreover, our experiment is more challenging because the teacher script preference data we generate is noisy (sampled from the BT model), whereas the other paper uses deterministic preference data.
>
> We have included our experiment in the last section of the paper PDF for your reference (Appendix F), and we plan to add more experiments before the camera-ready version. Due to time constraints, we were only able to include this experiment for now.
>
> [5] Meta-Reward-Net: Implicitly Differentiable Reward Learning for Preference-based Reinforcement Learning. 2022.
>
>
> Thank you once again for your feedback. We hope our responses have addressed your concerns and sincerely appreciate your consideration.

---

### Official Review · Reviewer_JdPq · 2025-10-31

**Soundness:** 2
**Presentation:** 3
**Contribution:** 2
**Rating:** 2
**Confidence:** 4

**Summary:**

The paper introduces Dual-Feedback Actor (DFA), a reinforcement learning algorithm that unifies scalar rewards and pairwise preferences into a single policy-update objective. The method directly models preference probabilities using policy log-probabilities without learning a separate reward model. The authors prove that, under a Bradley–Terry preference assumption on the soft-optimal Q-function, minimizing the DFA preference loss recovers the entropy-regularized Soft Actor-Critic (SAC) policy. Experiments demonstrate that DFA matches or exceeds SAC performance with more stable training on six MuJoCo tasks and outperforms reward-modeling baselines in a stochastic GridWorld with synthetic human feedback.

**Strengths:**

Strengths:
- The method fuses scalar rewards and pairwise preferences into one loss and update rule without requiring a separate reward model. This problem is important for the related field.
- Theorem 5.2 establishes that minimizing DFA’s state-wise preference loss recovers the SAC policy under Bradley–Terry preferences on the optimal Q-function.
- Support for off-policy training with replay buffers.

**Weaknesses:**

Weakness:
- There is no real human feedback experiments. I strongly suggest that the authors to conduct studies with real human feedback with substantial subjects. If the method aims to use human feedback to improve the system, but no experiment is conducted on real human feedback, and an insufficient number of individuals are used to demonstrate generalizability. It's difficult to convince the audience that the method is an effective approach for leveraging human feedback without human or with a limited number of subjects.
- Similarly, the theoretical guarantee relies on preferences exactly matching soft-optimal Q-value comparisons. This again limits the applicability to real noisy feedback.
- Preference synthesis uses nearest-neighbor Q-comparison, which may introduce bias and lacks ablations or robustness analysis to study this part.
- The experiments are not conducted in other environments, which leads to insufficient baseline comparisons with existing methods.
	- Some other environments to consider:
		- MetaDrive: Composing Diverse Driving Scenarios for Generalizable Reinforcement Learning (with interface to collect data.)
		- CREW: Facilitating Human-AI Teaming Research. GUIDE: Real-Time Human-Shaped Agents. (with human data provided and human interface).
- Nearest-state lookup in the replay buffer may scale poorly. The authors acknowledge the cost but do not quantify it.
- MuJoCo comparison only includes SAC; additional recent off-policy RL baselines (e.g., TD3, SPOT, or latent preference RL methods) would strengthen the claims.

**Questions:**

Please address the weakness points.

---

> ### Author Response · Authors · 2025-11-23
>
> We thank the reviewer for their valuable comments and address them below:
> >*There is no real human feedback experiments. I strongly suggest that...*
>
> We appreciate the reviewer’s comment on evaluating with real human feedback. In this work, we followed the standard RLHF evaluation protocol for continuous control by simulating preferences (via a Bradley--Terry model with controlled noise), which is common practice for ensuring reproducibility.
>      We agree that large-scale real human-in-the-loop studies are valuable future work. However, running such studies with substantial subject pools requires dedicated budget, logistics, and (often) IRB/ethics approvals, which are beyond the scope of the present submission. Importantly, our theoretical guarantees are agnostic to whether preferences are human- or synthetically generated. The results hinge on the preference model, not on the data collection modality. Methodologically, DFA natively supports both state-wise and trajectory-wise human labels without any modification to the loss or training loop.
>
> -----------
> >*Similarly, the theoretical guarantee relies on preferences ...*
>
> Our theory uses $Q^\star$ to characterize the clean fixed point, but the practical algorithm indeed synthesizes preferences from the current, noisy critic $Q_k$. In principle, one can extend the analysis to noisy $Q$ by assuming a concrete noise model for the critic, but the right model is not obvious here as critic errors in deep off-policy RL are often nonstationary and state--action dependent. Empirically, our experiments already operate in the regime described by the reviewer; $Q$ is far from optimal for much of training (i.e., it is a noisy $Q_k$), yet DFA trains stably and often more smoothly than SAC and reward-modeling baselines. This suggests the method is robust in practice to imperfect critics.
>
> -------------
> >*Preference synthesis uses nearest-neighbor Q-comparison...*
>
> If by bias, the reviewer means the noisy estimate of the Q function,  our experiments already operate with a non-converged and often suboptimal Q-function early in training (i.e., a noisy $Q_k$). Despite this, DFA trains stably and often more smoothly than SAC and reward-modeling baselines. This suggests that the method is practically robust to imperfect critics.
>      We have already conducted experiments other than nearest-neighbor, but they did not work well. For example, an alternative way to nearest-neighbor is sampling multiple actions from the \emph{current} policy at $s_i$. However, we observed this approach produced very similar actions, leading to small margins $|\Delta Q_k(s_i, a)|$ and weak gradients; this did not improve the policy reliably (we will share the results in the revisions, but they are just not trained policy). Our NN approach selects $s'_i$ as the closest state to $s_i$ in the buffer and uses its associated action $a'_i$. We then evaluate both candidates at the same target state $s_i$. This increases action diversity (since buffer actions come from earlier behavior policies) while controlling for state mismatch, and in practice yields larger $|\Delta Q_k|$ and more stable updates.
>
> ------------
>
> >*The experiments are not conducted in other environments, which leads to ...*
>
> Our submission already evaluates DFA on six standard MuJoCo control tasks (Walker2d, Hopper, Swimmer, Humanoid, InvertedPendulum, and MountainCarContinuous) and on multiple stochastic GridWorld settings that vary grid size, annotator pool size, and action-flip dynamics; more challenging GridWorld variants and complex MuJoCo environment (Humanoid) under preference supervision were provided in the appendix C. To further extend the empirical scope beyond locomotion and grid settings, we are currently running additional experiments on manipulation tasks from the Meta-World benchmark under both reward- and preference supervision. We will include these results as soon as they are available.
>
>
> -----------
>
> >*Nearest-state lookup in the replay buffer may scale poorly...*
>
> As noted in Appendix C, DFA incurred modest per-step overhead relative to SAC (e.g., Pendulum: $\sim$8h vs.\ $\sim$6.5h; Swimmer: $\sim$9h vs.\ $\sim$8h for 10M interactions), but often required fewer steps to reach the same return in some environments.

---

> > ### Author Response · Authors · 2025-11-23
> >
> > >*MuJoCo comparison only includes SAC; additional recent off-policy RL baselines ...*
> >
> > As stated in the paper, we initially focused on SAC because (i) extensive public benchmarks (e.g., \href{https://spinningup.openai.com/en/latest/spinningup/bench.html#benchmarks-for-spinning-up-implementations}{OpenAI SpinningUp}) consistently find SAC to be a top off-policy baseline on these continuous-control tasks (often matching or outperforming TD3 under comparable budgets), and (ii) our theory explicitly recovers the SAC policy under the Bradley--Terry preference model, making SAC the most directly relevant and strongest reward-based point of comparison. Also, we already include a Humanoid comparison with additional methods in Appendix C. That said, we agree that additional off-policy baselines would strengthen the empirical picture. In the revised version, we will add more methods across the six MuJoCo tasks.
> >
> > -----------
> >
> >
> > Thank you once again for your feedback. We hope our responses have addressed your concerns and sincerely appreciate your consideration. If there are any additional questions or points that require clarification, please do not hesitate to let us know.

---

> ### Author Response · Authors · 2025-12-03
>
> Dear Reviewer,
>
> As promised, aside from the previous experiments, we conducted more experiments on the Meta-World Hammer task in a new environment, using settings similar to those in the referenced paper [5]. Compared to the results reported there, our method outperforms other PbRL approaches and achieves performance very close to SAC. Moreover, our experiment is more challenging because the teacher script preference data we generate is noisy (sampled from the BT model), whereas the other paper uses deterministic preference data.
>
> We have included our experiment in the last section of the paper PDF for your reference (Appendix F), and we plan to add more experiments before the camera-ready version. Due to time constraints, we were only able to include this experiment for now.
>
> [5] Meta-Reward-Net: Implicitly Differentiable Reward Learning for Preference-based Reinforcement Learning. 2022.
>
>
> Thank you once again for your feedback. We hope our responses have addressed your concerns and sincerely appreciate your consideration.

---

### Official Review · Reviewer_9sYc · 2025-11-01

**Soundness:** 3
**Presentation:** 3
**Contribution:** 2
**Rating:** 4
**Confidence:** 3

**Summary:**

The paper introduces the DFA, an RL algorithm designed to learn from both scalar rewards and pairwise preferences. DFA's core mechanism avoids a separate reward-modeling step, which is common in many RLHF methods. Instead, it models the preference probability directly using the policy's log-probabilities.

When numerical rewards are available, DFA uses them to learn Q-values and then synthesizes preference pairs online by comparing Q-values of actions in the replay buffer. The paper provides a theoretical justification (Theorem 5.2) proving that, under a Bradley-Terry preference model based on the optimal Q-function ($Q^*$), minimizing DFA's preference loss recovers the optimal entropy-regularized SAC policy.

Empirically, the paper presents two main results (1) DFA (using its synthesized preferences) is shown to match or exceed SAC on six MuJoCo control tasks, exhibiting more stable training (2) DFA (using simulated human preferences) is shown to outperform RLHF baselines like RM+PPO and ZPG on a stochastic GridWorld.

**Strengths:**

- The paper proves that minimizing the DFA preference loss (under a BT assumption on $Q^*$) is equivalent to finding the optimal policy for the entropy-regularized SAC objective
- Section 6.2 provides a convincing experiment demonstrating DFA's capabilities in a stochastic MDP where only preference feedback is available

**Weaknesses:**

- The claims of "matching or exceeding SAC" rest on an algorithm that is confounded by a heuristic: a nearest-neighbor state search in the replay buffer to find a comparison. This heuristic is computationally expensive (a $k$-NN search on the buffer per gradient step). What would the training curves look like when plotting against the wall clock?
- The practical algorithm (Section 4.2) relies on synthesizing preferences from the current, noisy Q-estimate, $Q_k$. The theory, however, relies on the optimal, noise-free $Q^*$. The paper fails to analyze the impact of this noise. A noisy $Q_k$ will lead to noisy, incorrect preference labels, which could influence the stability during training.
- The paper could benefit from a wider range of experiments
- The state-wise preference (or action preference) setting is not natural in some settings (e.g., in robot learning, it is hard for a human to state whether a specific torque is better than another). The paper could benefit from a more thorough discussion of the per-trajectory preference as well as experimental results for this setting

**Questions:**

- The "preference synthesis" in 4.2 uses a nearest-neighbor state $s_i'$ to find the second action $a_i'$. What is the justification for this? Have the authors experimented with simpler, cheaper alternatives (e.g., sampling a random action $a_i'$ from the buffer, or sampling $a_i' \sim \pi_{\theta}( \cdot | s_i)$)?
- How does the noise from a non-converged Q-function affect the synthesized preference labels?
- Regarding the hyperparameters for Experiment 6.1, was $\alpha$ tuned individually for each environment, or was a single value used? Table 2 in the appendix seems to suggest it was tuned per-environment (e.g., 0.3 for Swimmer, 0.4 for MountainCarC). Can the authors confirm this? Given the sensitivity to $\alpha$ as shown in Figure 3 (b), it would seem that the convergence analysis should take the hyperparameter tuning into consideration when comparing with RM_2, for example. Similarly the the first point in "Weaknesses", it would be interesting to see a comparison with wall-clock time on the x-axis.

---

> ### Author Response · Authors · 2025-11-23
>
> We thank the reviewer for their valuable comments and address them below:
> >*The claims of "matching or exceeding SAC" rest on an algorithm that is confounded by a*
>
> Regarding wall-clock comparisons, as noted in the appendix C, DFA incurred modest per-step overhead relative to SAC (e.g., Pendulum: $\sim$8h vs.\ $\sim$6.5h; Swimmer: $\sim$9h vs.\ $\sim$8h for 10M interactions), but often required fewer steps to reach the same return in some environments. In the revised version, we will include these wall-clock curves alongside the existing system-probe plots.
>
> -----
> > *The practical algorithm (Section 4.2) relies on synthesizing preferences...*
>
> As noted by the reviewer, our theory uses $Q^\star$ to characterize the clean fixed point, but the practical algorithm indeed synthesizes preferences from the current, noisy critic $Q_k$. In principle, one may extend the analysis to noisy $Q$ but the right noise model is not obvious here, as critic errors in deep off-policy RL are often nonstationary, and state--action dependent. Empirically, our experiments already operate in the regime where $Q$ is far from optimal for much of training (i.e., it is a noisy $Q_k$), yet DFA trains stably and often more smoothly than SAC and reward-modeling baselines. This suggests the method is robust in practice to imperfect critics.
>
> -------
>
> >*The paper could benefit from a wider range of experiments*
>
> We appreciate the suggestion to broaden the experimental evaluation. Our submission already evaluates DFA on six standard MuJoCo control tasks (Walker2d, Hopper, Swimmer, Humanoid, InvertedPendulum, and MountainCarContinuous) and on multiple stochastic GridWorld settings that vary grid size, annotator pool size, and action-flip dynamics; more challenging GridWorld variants and complex MuJoCo environment (Humanoid) under preference supervision were provided in the appendix C. To further extend the empirical scope beyond locomotion and grid settings, we are currently running additional experiments on manipulation tasks from the Meta-World benchmark under both reward and preference supervision; we will include these results as soon as they are available.
>
> ---------
>
> >*The state-wise preference (or action preference) setting is not natural...*
>
>  We agree that in some robotic settings, raw torques are not a natural unit for human comparison, and that trajectory-level judgments can be more intuitive. At the same time, state-wise feedback is practical in some applications: e.g., when a mobile robot is stuck at a corner or blocked by an obstacle, a human can quickly indicate a preferred direction (turn left vs. reverse) as a pairwise action choice; interactive navigation, teleoperation, and TAMER-style coaching (a human-in-the-loop training where the trainer “coaches” by giving quick positive/negative feedback as the agent acts) provide state-level feedback [1][2]. Importantly, DFA natively supports trajectory-level feedback. Section 4 defines the trajectory loss, Appendix B connects it to the state-wise formulation, and our GridWorld study in Section 6 already uses trajectory-pair supervision; more experiments of this type are already in Appendix C. In the revised version, we will expand the discussion of when to prefer state-wise vs.\ trajectory-wise feedback, and include additional trajectory-level experiments (including in Meta-World manipulation). Finally, when numeric rewards exist, DFA can synthesize state-wise preferences off-policy from the replay buffer, reducing human burden where per-trajectory annotation is costly.
>
> >[1]: TAMER: Training an Agent Manually via Evaluative Reinforcement:Knox, W. Bradley and Stone, Peter
>
> >[2] Deep TAMER: Interactive Agent Shaping in High-Dimensional State Spaces: Warnell, Garrett and Waytowich, Nicholas and Lawhern, Vernon and Stone, Peter.
> -------------
> >*The "preference synthesis" in 4.2 uses a nearest-neighbor state ....*
>
> We have already conducted experiments with other strategies that were mentioned, but they did not work well. Sampling multiple actions from the \emph{current} policy at $s_i$ produced very similar actions, leading to small margins $|\Delta Q_k(s_i, a)|$ and weak gradients; this did not improve the policy reliably (we will share the results in the revisions, but they are just not trained policy). Our NN approach selects $s'_i$ as the closest state to $s_i$ in the buffer and uses its associated action $a'_i$. We then evaluate both candidates at the same target state $s_i$. This increases action diversity (since buffer actions come from earlier behavior policies) while controlling for state mismatch, and in practice yields larger $|\Delta Q_k|$ and more stable updates.

---

> > ### Author Response · Authors · 2025-11-23
> >
> > ------------
> > >*How does the noise from a non-converged Q-function affect the synthesized preference labels?*
> >
> > As noted above, our experiments already operate with a non-converged and often suboptimal Q-function early in training (i.e., a noisy $Q_k$). Despite this, DFA trains stably and often more smoothly than SAC and reward-modeling baselines. This suggests that the method is practically robust to imperfect critics.
> >
> > -------------
> > >* Regarding the hyperparameters for Experiment 6.1, was tuned individually for each environment, or was a single value used?*
> >
> >
> > Yes, we tuned the DFA temperature $\alpha$ per environment; Table~2 reports the selected values (e.g., $0.3$ for Swimmer, $0.4$ for MountainCarContinuous). We applied comparable hyperparameter tuning effort to all baselines, including $RM_2$: for RM methods we adjusted reward-model capacity (e.g., MLP width/depth), the number of reward-model pretraining iterations, and PPO settings. We then reported, for each method, the best-performing configuration to ensure a fair comparison. We will make this explicit in the main text and include a summary table of the sweep ranges. Regarding wall-clock comparisons, as noted in the appendix, DFA incurred modest per-step overhead relative to SAC (e.g., Pendulum: $\sim$8h vs.\ $\sim$6.5h; Swimmer: $\sim$9h vs.\ $\sim$8h for 10M interactions), but often required fewer steps to reach the same return in some environments. In the revised version, we will include these wall-clock curves alongside the existing system-probe plots.
> >
> > -------------
> >
> >
> >
> > Thank you once again for your feedback. We hope our responses have addressed your concerns and sincerely appreciate your consideration. If there are any additional questions or points that require clarification, please do not hesitate to let us know.

---

> ### Author Response · Authors · 2025-12-03
>
> Dear Reviewer,
>
> As promised, aside from the previous experiments, we conducted more experiments on the Meta-World Hammer task in a new environment, using settings similar to those in the referenced paper [5]. Compared to the results reported there, our method outperforms other PbRL approaches and achieves performance very close to SAC. Moreover, our experiment is more challenging because the teacher script preference data we generate is noisy (sampled from the BT model), whereas the other paper uses deterministic preference data.
>
> We have included our experiment in the last section of the paper PDF for your reference (Appendix F), and we plan to add more experiments before the camera-ready version. Due to time constraints, we were only able to include this experiment for now.
>
> [5] Meta-Reward-Net: Implicitly Differentiable Reward Learning for Preference-based Reinforcement Learning. 2022.
>
>
> Thank you once again for your feedback. We hope our responses have addressed your concerns and sincerely appreciate your consideration.

---

### Meta-Review · Area_Chair_Bkfw · 2026-01-10

**Summary:**

The paper proposes a unified framework for combining scalar rewards and pairwise preferences in reinforcement learning, with a clear theoretical connection to entropy-regularized SAC under idealized assumptions. Reviewers acknowledge the conceptual simplicity of the formulation and the effort to bridge reward-based and preference-based learning.

However, several core concerns remain unresolved. The empirical evidence is insufficient to fully substantiate the main claims: evaluations rely heavily on comparisons to SAC with limited baseline diversity, and key settings such as mixed reward–preference supervision and robustness to noisy or inconsistent preferences are not directly tested. The reliance on synthesized preferences from a noisy critic introduces a theory–practice gap that is not formally analyzed or empirically isolated. Additional concerns were raised regarding computational overhead, scalability, and the lack of real human-feedback studies.

Overall, reviews express mixed opinions with marginal scores and no explicit post-rebuttal confirmations of resolved major issues. Given the interrupted review process, this assessment reflects a best-effort, conservative judgment based solely on the available reviews and discussion, without assuming any score changes.

**Reviewer Concerns:**

Reviewer 9sYc:

Addressed:
• Hyperparameter tuning clarification (α): authors confirm per-environment tuning.

Partially addressed:
• Computational cost of nearest-neighbor preference synthesis: limited wall-clock numbers provided, but no full scaling analysis.
• Experimental breadth and trajectory-level preferences: minor additions and discussion, but limited new evidence.

Outstanding:
• Theory–practice gap (use of noisy Qk vs. theoretical Q*): no formal analysis or targeted experiments quantifying the impact.

Reviewer JdPq:

Addressed:
• None.

Partially addressed:
• Computational overhead and NN preference synthesis: qualitative explanation and limited timing results only.

Outstanding:
• Absence of real human-feedback experiments.
• Reliance on idealized preference assumptions with no robustness analysis.
• Insufficient baseline and environment coverage beyond SAC.

Reviewer Gbv4:

Addressed:
• Clarification of preference synthesis pipeline (Q-values trained from environment rewards).
• Clarification of preference label counts.

Partially addressed:
• Motivation for dual feedback: narrative explanation without direct experimental validation.

Outstanding:
• Lack of strong PbRL baselines and diverse/challenging environments.
• Limited engagement with PbRL/robotics-focused related work.

Reviewer 2Mgv:

Addressed:
• Inclusion/clarification of an OnlineDPO baseline.

Partially addressed:
• Additional environments and robustness discussion: limited new evidence.

Outstanding:
• No direct mixed-signal (reward + preference) experiment.
• Missing additional RL baselines (e.g., PPO/TD3/Rainbow) and IPL.
• No targeted evaluation under noisy/inconsistent preferences.

**Reviewer Scores:**

Reviewer 9sYc:

Original score: 4

Likely post-rebuttal score: 4

Justification:
• No explicit reviewer signal indicating satisfaction or intent to raise the score.
• Outstanding major concern on theory–practice mismatch and limited cost analysis.

Reviewer JdPq:

Original score: 2

Likely post-rebuttal score: 2

Justification:
• No explicit reviewer signal indicating a score change.
• Multiple major concerns remain unresolved (human feedback, theory robustness, baselines).

Reviewer Gbv4:

Original score: 2

Likely post-rebuttal score: 2

Justification:
• No explicit reviewer signal indicating a score change.
• Core experimental and baseline concerns remain outstanding.

Reviewer 2Mgv:

Original score: 4

Likely post-rebuttal score: 4

Justification:
• Reviewer expressed conditional willingness to increase the score, but provided no confirmation post-rebuttal.
• Major experimental gaps (mixed-signal setting, baselines, robustness) remain unresolved.

---

### Decision · Program_Chairs · 2026-01-26

Reject